# Optimization without retraction on the random generalized Stiefel manifold for canonical correlation analysis

## Abstract

Optimization over the set of matrices that satisfy $X^\top BX = I_p$, referred to as the generalized Stiefel manifold, appears in many applications such as canonical correlation analysis (CCA) and the generalized eigenvalue problem. Solving these problems for large-scale datasets is computationally expensive and is typically done by either computing the closed-form solution with subsampled data or by iterative methods such as Riemannian approaches. Building on the work of Ablin & Peyré (2022), we propose an inexpensive iterative method that does not enforce the constraint in every iteration exactly, but instead it produces iterations that converge to the generalized Stiefel manifold. We also tackle the random case, where the matrix $B$ is an expectation. Our method requires only matrix multiplications and has the same sublinear convergence rate as its Riemannian counterpart. Experiments demonstrate its effectiveness in various machine learning applications involving generalized orthogonality constraints, including CCA for measuring model representation similarity.

## 1 Introduction

Many problems in machine learning and engineering, including the canonical correlation analysis (CCA) (Hotelling, 1936), linear discriminant analysis (LDA) (McLachlan, 1992), and the generalized eigenvalue problem (GEVP) (Saad, 2011), can be formulated as the following optimization problem

$$\min f(X) := \mathbb{E}[f_\xi(X)], \text{ s.t. } X \in \mathrm{St}_B(p, n) := \left\{ X \in \mathbb{R}^{n \times p} | X^\top BX = I_p \right\} \text{ and } B = \mathbb{E}[B_\zeta], \tag{1}$$

where the objective function $f$ is the expectation of $L$-smooth functions $f_\xi$, and $B \in \mathbb{R}^{n \times n}$ is a positive definite matrix defined as the expectation $B = \mathbb{E}_\zeta[B_\zeta] \succ 0$, and $\xi, \zeta$ are independent random variables. We only assume that individual random matrices $B_\zeta$ are positive semi-definite.

The feasible set $\mathrm{St}_B(p, n) \subset \mathbb{R}^{n \times p}$ defines a smooth manifold referred to as the *generalized Stiefel manifold*, and for noiseless $B$, the optimization problem can be solved by Riemannian techniques (Absil et al., 2008; Boumal, 2023). Riemannian methods produce a sequence of iterates belonging to the set $\mathrm{St}_B(p, n)$ by performing *retractions*, which are projections on the constraint that are accurate up to the first-order and in the case of $\mathrm{St}_B(p, n)$ require non-trivial linear algebra operations such as eigenvalue or Cholesky decomposition. In contrast, infeasible approaches, such as the augmented Lagrangian method, are typically employed in deterministic setting when the constraint set does not have a convenient projection, e.g. by the lack of a closed-form expression or because they require solving an expensive optimization problem themselves (Bertsekas, 1982). Infeasible approaches produce iterates do not strictly remain on the constraint but gradually converge to the feasible set by solving a sequence of unconstrained optimization problems. However, solving the optimization subproblems in each iteration might be computationally expensive and the methods are sensitive to the choice of hyper-parameters, both in theory and in practice.

In this paper, we consider the setting (1) where the constraint itself is *stochastic*, i.e. the matrix $B$ is an expectation, for which, neither Riemannian methods nor infeasible optimization techniques, are well-suited. In particular, we are interested in the case where we only have access to i.i.d. samples from $\xi$ and $\zeta$, and not to the full function $f$ and matrix $B$.

We design an iterative *landing* method requiring only matrix multiplications that provably converges to a critical point of (1) under stochastic constraints. The main principle of the method is depicted in the diagram in Figure 1 and is inspired by the recent line of work for the deterministic constraint of the orthogonal set (Ablin & Peyré, 2022), and the *Stiefel manifold* (Gao et al., 2022b; Ablin et al., 2023; Schechtman et al., 2023). Instead of performing projections after each iteration, the proposed algorithm only

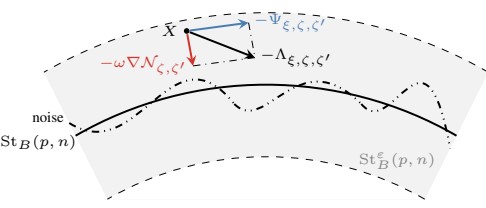

Figure 1: Illustration of the landing field.

tracks an approximate distance to the constraint and remains within an initially prescribed $\varepsilon$-safe region and finally "lands" on, i.e. converges to, the manifold by following an unbiased estimator of the direction towards the manifold.

The stochastic landing iteration for solving (1) is a simple, cheap, and stochastic update rule

$$X^{k+1} = X^k - \eta_k \Lambda_{\xi^k, \zeta^k, \zeta'^k}(X^k) \quad \text{with} \quad \Lambda_{\xi, \zeta, \zeta'}(X) = \Psi_{\xi, \zeta, \zeta'}(X) + \omega \nabla \mathcal{N}_{\zeta, \zeta'}(X), \quad (2)$$

whose two components are

$$\Psi_{\xi, \zeta, \zeta'}(X) = 2 \,\text{skew}\left(\nabla f_\xi(X) X^\top B_\zeta\right) B_{\zeta'} X \quad \text{and} \quad \nabla \mathcal{N}_{\zeta, \zeta'}(X) = 2 B_\zeta X \left(X^\top B_{\zeta'} X - I_p\right), \quad (3)$$

where $\nabla \mathcal{N}_{\zeta, \zeta'}(X)$ is an unbiased stochastic estimator of the gradient of $\mathcal{N}(X) = \frac{1}{2}\|X^\top B X - I_p\|_F^2$ and $\text{skew}(A) = (A - A^\top)/2$. The above formula (2) is the more general formula of the landing field in the case where both the function $f$ and the constraint matrix $B$ are stochastic; the deterministic case is recovered by simply putting $\nabla f_\xi = \nabla f$ and $B_\zeta = B_{\zeta'} = B$ in the formula. Note that in many applications of interest, $B_\zeta = \sum_{i=1}^r x_i x_i^\top / r$ is a subsampled covariance matrix with batch-size $r$, that is of rank $r$ when $r \leq n$. The landing method benefits in this setting since the cost of multiplication by $B_\zeta$, which is the dominant cost of (3) becomes $\mathcal{O}(npr)$ instead of $\mathcal{O}(n^2 p)$ where $r$ is the batch size. The landing method never requires to form the matrix $B$, thus having space complexity defined by only saving the iterates: $\mathcal{O}(np)$ instead of $\mathcal{O}(n^2)$.

We demonstrate that the iteration converges with fixed step size in the deterministic case (Theorem 2.7) and with decaying step size in the stochastic case (Theorem 2.8), with a rate that matches that of stochastic Riemannian gradient descent on $\text{St}_B(p, n)$. The advantages of the landing field in (2) are that i) its computation involves only parallelizable matrix multiplications, which is cheaper than the computations of the Riemannian gradient and retraction and ii) it handles gracefully the stochastic constraint, while Riemannian approaches need to estimate the constraint $B$.

While the presented theory holds for a general smooth, possibly non-convex objective $f$, a particular problem that can be either solved by (1) or framed as an optimization over the product manifold of two $\text{St}_B(p, n)$ is CCA, which is a widely used technique for measuring similarity between datasets (Raghu et al., 2017). CCA aims to find the top-$p$ most correlated principal components $X, Y \in \mathbb{R}^{n \times p}$, for two zero-centered datasets $D_1 = (d_1^1, \ldots, d_1^N), D_2 = (d_2^1, \ldots, d_2^N) \in \mathbb{R}^{n \times N}$ of $N$ iid samples from two different distributions and is formulated as

$$\min_{X, Y \in \mathbb{R}^{n \times p}} \mathbb{E}_i \left[ -\text{Tr}(X^\top d_1^i (d_2^i)^\top Y) \right] \text{ s.t. } X^\top \mathbb{E}_i[d_1^i (d_1^i)^\top] X = I_p \text{ and } Y^\top \mathbb{E}_i[d_2^i (d_2^i)^\top] Y = I_p, \quad (4)$$

where the expectations are w.r.t. the uniform distribution over $\{1, \ldots, N\}$. Here, the constraint matrices $B_\zeta$ correspond to individual or mini-batch sample covariances, and the constraint is that the large matrix $Z = (X^\top, Y^\top)^\top$ is in the generalized Stiefel manifold.

The following subsection gives a brief overview on the current optimization techniques for solving (1) and its forthcoming generalization (5). The rest of the paper is organized as follows.

- In Section 2 we give a form to a generalized landing algorithm for solving a smooth optimization problem $\min_{x \in \mathcal{M}} f(x)$ on a smooth manifold $\mathcal{M}$ (5), which under suitable assumptions, converges to a critical point with the same sublinear rate $\mathcal{O}(1/K)$, where $K$ is the iteration number, as its Riemannian counterpart (Boumal et al., 2019), see Theorem 2.7. Unlike previous works, our analysis is based on a smooth merit function allowing us to obtain a convergence result for the stochastic variant of the algorithm, when having an unbiased estimator for the landing field, see Theorem 2.8

- In Section 3 we build on the general theory developed in the previous section and prove that the update rule in (2) converges to a critical point of (1) both in the deterministic case with the rate $\mathcal{O}(1/K)$ and in expectation with the rate $\mathcal{O}(1/\sqrt{K})$ in the case when both the gradient of the objective function and the constraint are stochastic estimates.

- In Section 4 we numerically demonstrate the efficiency of the proposed method on a deterministic example of solving a generalized eigenvalue problem and the stochastic CCA.

**Notation.** We denote vectors by lower case letters $x, y, z, \ldots$, matrices with uppercase letters $X, Y, Z, \ldots$, and $I_n$ denotes the $n \times n$ identity matrix. Let $\mathrm{D}f(x)[v] = \lim_{t \to 0} (f(x + tv) - f(x))/t$ denote the derivative of $f$ at $x$ along $v$. The $\| \cdot \|$ denotes the $\ell_2$-norm or the Frobenius norm for matrices, whereas $\| \cdot \|_2$ denotes the operator norm induced by $\ell_2$-norm.

### 1.1 PRIOR WORK RELATED TO THE OPTIMIZATION ON THE GENERALIZED STIEFEL MANIFOLD

**Riemannian optimization.** A widely used approach to solving optimization problems constrained to manifold as in (5) are the techniques from Riemannian optimization. These methods are based on the observation that smooth sets can be locally approximated by a linear subspace, which allows to extend classical Euclidean optimization methods, such as gradient descent and the stochastic gradient descent to the Riemannian setting. For example, Riemannian gradient descent iterates $x^{k+1} = \mathrm{Retr}_{\mathcal{M}}(x^k, -\eta_k \mathrm{grad} f(x^k))$, where $\eta_k > 0$ is the stepsize at iteration $k$, $\mathrm{grad} f(x^k)$ is the Riemannian gradient that is computed as a projection of $\nabla f(x^k)$ on the tangent space of $\mathcal{M}$ at $x^k$, and $\mathrm{Retr}$ is the *retraction* operation, which projects the updated iterate along the direction $-\eta_k \mathrm{grad} f(x^k)$ on the manifold and is accurate up to the first-order, i.e., $\mathrm{Retr}_{\mathcal{M}}(x, d) = x + d + o(\|d\|)$. Retractions allow the implementation of Riemannian counterparts to classical Euclidean methods on the generalized Stiefel manifold, such as Riemannian (stochastic) gradient descent, trust-region methods (Absil et al., 2007), and accelerated methods (Ahn & Sra, 2020); for an overview, see (Absil et al., 2008; Boumal, 2023).

There are several ways to compute a retraction to the generalized Stiefel manifold, which we summarize in Table 1 and we give a more detailed explanation in Appendix A. Overall, we see that the landing field (3) is much cheaper to compute than all these retractions in two cases: i) when $n \simeq p$, then the bottleneck in the retractions becomes the matrix factorization, which, although they are of the same complexity as matrix multiplications, are much more expensive and hard to parallelize, ii) when $n$ gets extremely large, the cost of all retractions grows quadratically with $n$, while the use of mini-batches of size $r$ allows computing the landing field in linear time. We show the practical cost of computing retractions in Fig 6b in the appendices.

**Infeasible optimization methods.** Infeasible methods, such as the augmented Lagrangian method, seek to solve a deterministic minimization problem with $\mathcal{L}(x, \lambda)$, such as the one introduced later in (9), by updating the solution vector $x$ and the vector of Lagrange multipliers $\lambda$ respectively (Bertsekas, 1982). This is typically done by solving a sequence of optimization problems of $\mathcal{L}(\cdot, \lambda_k)$ followed by a first-order update of the multipliers $\lambda_{k+1} = \lambda_k - \beta h(x_k)$ depending on the penalty parameter $\beta$. The iterates are gradually pushed towards the constraint by increasing the penalty parameter $\beta$. However, each optimization problem might be expensive, and the methods are sensitive to the correct choice of the penalty parameter $\beta$.

Recently, a number of works explored the possibility of infeasible methods for optimization on Riemannian manifolds in order to eliminate the cost of retractions, which can be limiting in some situations, e.g. when evaluation of stochastic gradients is cheap. The works of (Gao et al., 2019a; 2022a) proposed a modified augmented Lagrangian method which allows for fast computation and better better bounds on the penalty parameter $\beta$. Ablin & Peyré (2022) designed a simple iterative method

| | Matrix factorizations | Deterministic complexity | Stochastic complexity |
|---|---|---|---|
| Polar (Yger et al., 2012) | matrix inverse square root | $\mathcal{O}(n^2 p)$ | - |
| SVD-based (Mishra & Sepulchre, 2016) | SVD | $\mathcal{O}(n^2 p)$ | - |
| Cholesky-QR (Sato & Aihara, 2019) | Cholesky, matrix inverse | $\mathcal{O}(n^2 p)$ | - |
| $\Lambda(X)$ formula in (2) | None | $\mathcal{O}(n^2 p)$ | $\mathcal{O}(nrp)$ |

Table 1: Costs of retractions on the generalized Stiefel manifold. The matrices are of size $n \times p$ with $p \leq n$, and $r$ is the rank of the stochastic matrices $B_\zeta$. Matrix factorizations are hard to parallelize.

called *landing*, consisting of two orthogonal components, to be used on the orthogonal group, which was later expanded to the Stiefel manifold (Gao et al., 2022b; Ablin et al., 2023). Schechtman et al. (2023) expanded the *landing* approach to be used on a general smooth constraint using a non-smooth merit function. More recently, Goyens et al. (2023) analysed the classical Fletcher's augmented Lagrangian for solving smoothly constrained problems through the Riemannian perspective and proposed an algorithm that provably finds second-order critical points of the minimization problem.

## 1.2 METHODS FOR THE GENERALIZED EIGENVALUE PROBLEM AND CCA.

**Deterministic methods.** A lot of effort has been spent in recent years on finding fast and memory-efficient solvers for CCA/GEVP. Majority of the existing methods for computing the top-$p$ vector problem aim to circumvent the need to compute $B^{-\frac{1}{2}}$ or $B^{-1}$, e.g. by using an approximate solver to compute the action of multiplying with $B^{-1}$. The classic Lanczos algorithm for computation of eigenvalues can be adapted to the GEVP by noting that we can look for standard eigenvectors of $B^{-1}A$, see (Saad, 2011, Algorithm 9.1). (Ma et al., 2015) proposes `AppGrad` which performs a projected gradient descent with $\ell_2$-regularization and proves its convergence when initialized sufficiently close to the minimum. The work of (Ge et al., 2016) proposes `GenELinK` algorithm based on the block power method, using inexact linear solvers, that has provable convergence with a rate depending on the eigenvalue gap $1/\delta$. Allen-Zhu & Li (2017) improves upon this in terms of the eigenvanlue gap and proposes the doubly accelerated method LazyEV, which is based on the shift-and-invert strategy with iteration complexity that depends on $1/\sqrt{\delta}$. Xu & Li (2020) present a first-order Riemannian algorithm that computes gradients using fast linear solvers to approximate the action of $B^{-1}$ and performs polar retraction. (Meng et al., 2021) presents a Riemannian optimization technique that finds top-$p$ vectors using online estimates of the covariance matrices with $\mathcal{O}(n^2p)$ per-iteration complexity with time complexity of $\mathcal{O}(1/K)$.

**Stochastic methods.** While the stochastic CCA problem is of high practical interest, fewer works consider it. Although several of the aforementioned deterministic solvers can be implemented for streaming data using sampled information (Ma et al., 2015; Wang et al., 2016; Meng et al., 2021), they do not analyse stochastic convergence. The main challenge comes from designing an unbiased estimator for the whitening part of the method that ensures the constraint $X^\top BX = I$ in expectation. Arora et al. (2017) propose a stochastic approximation algorithm, MSG, that keeps a running weighted average of covariance matrices used for projection, requiring computing $B^{-1/2}$ at each iteration. Additionally, the work of (Gao et al., 2019b) proves stochastic convergence of an algorithm based on the shift-and-invert scheme and SVRG to solve linear subproblems, but only for the top-1 setting.

**Comparison with the landing.** Constrained optimization methods such as the augmented Lagrangian methods and Riemannian optimization techniques can be applied on stochastic problems only when the gradient of the objective function is random, however, not on problems with *stochastic constraints*. The landing method has provable global convergence guarantees with the same asymptotic rate as its Riemannian counterpart, while also allowing for stochasticity in the constraint. Our

| | Stochastic | Matrix factorizations | Total operation count complexity for e-stationarity | Memory |
|---|:---:|:---:|:---:|:---:|
| AppGrad[*] (Ma et al., 2015) | - | SVD | $\mathcal{O}((n^2p\kappa_B + p^2n)\delta^{-1}\log(1/e) + Nn^2)$ | $n^2$ |
| CCALin (Ge et al., 2016) | - | linear solver | $\mathcal{O}((n^2p\sqrt{\kappa_B} + p^2n)\delta^{-1}\log(1/e) + Nn^2)$ | $n^2$ |
| rgCCALin (Xu & Li, 2020) | - | linear solver | $\mathcal{O}((n^2p\sqrt{\kappa_B} + p^2n)\delta'^{-2}\log(1/e) + Nn^2)$ | $n^2$ |
| LazyCCA (Allen-Zhu & Li, 2017) | - | linear solver | $\mathcal{O}((n^2p\sqrt{\kappa_B} + p^2n)\delta^{-1/2}\log(1/e) + Nn^2)$ | $n^2$ |
| MSG (Arora et al., 2017) | ✓ | inverse square root | $\mathcal{O}(n^3(p(\sigma')^2 + p^2\kappa_B^2)/e^2)$ | $n^2$ |
| $\Lambda(X)$ formula[†] in (3) | ✓ | None | $\mathcal{O}(\kappa_B^5\sigma^2np/e^2)$ | $np$ |

Table 2: Overview of CCA and GEVP solvers for finding top-$p$ vectors simultaneously that achieve $e$-stationary point, i.e. $\|\nabla f(X^k)\| \leq e$. We assume that the number of samples is much greater than the dimension $N \gg n$. Deterministic methods depend on the gap $\delta = \beta_p - \beta_{p+1}$, while stochastic methods are independent of the $\delta$ and depend on the variance, where $\sigma'$ is the variance of the data $x$, whereas $\sigma$ is the variance of the covariance estimate $xx^\top$. The first three methods achieve linear rate $\mathcal{O}(\log(1/e))$, while the last two methods have sublinear rate $\mathcal{O}(1/e^2)$. "Stochastic" marks methods with convergence analysis for the stochastic case. Deterministic methods require forming the matrix $B$ at the start with additional cost $\mathcal{O}(Nn^2)$. [*]marks local convergence result to the minimum and [†]marks convergence to a critical point.

work is conceptually related to the recently developed infeasible methods (Ablin & Peyré, 2022; Ablin et al., 2023; Schechtman et al., 2023), with the key difference of constructing a smooth merit function for a general constraint $h(x)$ that enables convergence analysis of iterative updates with error in the normal space of $\mathcal{M}$. In Table 2 we show the overview of relevant GEVP/CCA methods by comparing their asymptotic operations cost required to converge to an $e$-critical point[1]. The operation count takes into account both the number of iterations and the per-iteration cost, which is bounded asymptotically for the landing in Proposition 3.4. Despite the landing iteration (3) being designed for a general non-convex smooth problem (1) and not being tailored specifically to GEVP/CCA, we achieve theoretically interesting rate, which outperforms the other methods for well-conditioned matrices, when $\kappa$ is small, and when the variance of samples is potentially small. Additionally, we provide an improved space complexity $\mathcal{O}(np)$ by not having to form the full matrix $B$ and only to save the iterates.

## 2 GENERALIZED LANDING WITH STOCHASTIC CONSTRAINTS

This section is devoted to analyzing the landing method in the general case where the constraint is given by the zero set of a smooth function. We will later use these results in Section 3 for the analysis on $\mathrm{St}_B(p,n)$. The theory presented here improves on that of Schechtman et al. (2023) in two important directions. First, we generalize the notion of relative descent direction, which allows us to consider a richer class than that of *geometry-aware orthogonal directions* (Schechtman et al., 2023, Eq.18). Second, we do not require any structure on the noise term $E$ in the stochastic case, while A2(iii) in (Schechtman et al., 2023) requires the noise to be in the tangent space. This enhancement is due to the smoothness of our merit function $\mathcal{L}$, while Schechtman et al. (2023) consider a non-smooth merit function. Importantly, for the case of $\mathrm{St}_B(p,n)$ with the formula given in (3), there is indeed noise in the normal space, rendering Schechtman et al. (2023)'s theory inapplicable, while we show in the next section that Theorem 2.8 applies in that case.

Given a continuously differentiable function $f : \mathbb{R}^d \to \mathbb{R}$, we solve the optimization problem:

$$\min_{x \in \mathbb{R}^d} f(x) \qquad \text{s.t.} \quad x \in \mathcal{M} = \left\{ x \in \mathbb{R}^d \, : \, h(x) = 0 \right\}, \tag{5}$$

where $h : \mathbb{R}^d \to \mathbb{R}^q$ is continuously differentiable, non-convex, $q \in \mathbb{N}$ represents the number of constraints, and $\mathcal{M}$ defines a smooth manifold set. We will consider algorithms that stay within an initially prescribed $\varepsilon$-proximity region

$$\mathcal{M}^\varepsilon = \left\{ x \in \mathbb{R}^d \, : \, \|h(x)\| \leq \varepsilon \right\}. \tag{6}$$

The first assumption we make is a blanket assumption from $f$ having a smooth derivative. The second one requires that the differential $\mathrm{D}h(x)^*$ inside the $\varepsilon$-safe region has bounded singular values.

**Assumption 2.1** (Smoothness of the objective). *The objective function $f : \mathbb{R}^d \to \mathbb{R}$ is continuously differentiable and its gradient is $L_f$-Lipschitz.*

**Assumption 2.2** (Smoothness of the constraint). *Let $\mathrm{D}h(x)^* : \mathbb{R}^q \to \mathbb{R}^d$ be the adjoint of the differential of the constraint function $h$. The adjoint of the differential has bounded singular values for $x$ in the safe $\varepsilon$-region, i.e., $\forall x \in \mathcal{M}^\varepsilon : \quad \bar{C}_h \leq \sigma\left(\mathrm{D}h(x)^*\right) \leq C_h$. Additionally, the gradient $\nabla \mathcal{N}(x)$ of the penalty term $\mathcal{N}(x) = \frac{1}{2}\|h(x)\|^2$ is Lipschitz continuous with constant $L_\mathcal{N}$ over $\mathcal{M}^\varepsilon$.*

Assumption 2.1 is standard in optimization. Assumption 2.2 is necessary for the analysis of smooth constrained optimization (Goyens et al., 2023) and holds, e.g., when $\mathcal{M}^\varepsilon$ is a compact set, $\mathrm{D}h(x)^*$ is smooth and the constraints defined by $h$ are independent. Next, we define a relative gradient descent direction $\Psi(x)$, which is an extension of the Riemannian gradient outside of the manifold.

**Definition 2.1** (Relative descent direction). *A relative descent direction $\Psi(x) : \mathbb{R}^d \to \mathbb{R}^d$, with a parameter $\rho > 0$ that may depend on $\varepsilon$ satisfies:*

*(i) $\forall x \in \mathcal{M}^\varepsilon, \quad \forall v \in \mathrm{span}(\mathrm{D}h(x)^*) : \langle \Psi(x), v \rangle = 0$;*

*(ii) $\forall x \in \mathcal{M}^\varepsilon$ we have that $\langle \Psi(x), \nabla f(x) \rangle \geq \rho \|\Psi(x)\|^2$;*

---

[1] Note that some of the works show *linear* convergence to a global minimizer, which by the smoothness of $f$ also implies a $e$-critical point, whereas we prove $1/e^2$ convergence to a critical point. For the purpose of the comparison, we overlook this difference. Also, there are no local non-global minimizers in the GEVP.

*(iii) For $x \in \mathcal{M}$, we have that $\langle \Psi(x), \nabla f(x) \rangle = 0$ if and only if $x$ is a critical point of $f$ on $\mathcal{M}$.*

In short, the *relative descent direction* must be orthogonal to the normal space $\text{span}(\mathrm{D}h(x)^*)$ while remaining positively aligned with the Euclidean gradient $\nabla f(x)$. While there may be many examples of relative descent directions, a particular example is the Riemannian gradient of $f$ with respect to the sheet manifold $h(x) = c$ when $\|c\| \leq \varepsilon$. Note, the above definition is not scale invariant to $\rho$, i.e. taking $c\Psi(x)$ for $c > 0$ will result in $c\rho$, and this is in line with the forthcoming convergence guarantees deriving upper bound on $\|\Psi(x)\|_F$.

**Proposition 2.2** (Riemannian gradient is a relative descent direction). *The Riemannian gradient of $f$ in respect to the sheet manifold $h(x) = c$, defined as*

$$\text{grad} f(x) = \nabla f(x) - \mathrm{D}h(x)^* \left(\mathrm{D}h(x)^*\right)^\dagger \nabla f(x), \tag{7}$$

*where $c \in \mathbb{R}^p$ is an error term such that $\|c\| \leq \varepsilon$, $\mathrm{D}h(x)$ denotes a differential, and $\mathrm{D}h(x)^* \left(\mathrm{D}h(x)^*\right)^\dagger$ acts as a projection on the normal space of $h(x) = c$ at $x$, qualifies as a descent direction on $\mathcal{M}^\varepsilon$ with $\rho = 1$.*

The proof can be found in the appendices in Subsection C.1. Such extension of the Riemannian gradient to the whole space was already considered by Gao et al. (2022b) in the particular case of the Stiefel manifold and by Schechtman et al. (2023). We now define the general form of the *deterministic* landing iteration as

$$x^{k+1} = x^k - \eta_k \Lambda(x^k) \quad \text{with} \quad \Lambda(x) = \Psi(x) + \omega \nabla \mathcal{N}(x), \tag{8}$$

where $\Psi(x)$ is a relative descent direction described in Def. 2.1, $\nabla \mathcal{N}(x)$ is the gradient of the penalty $\mathcal{N}(x) = \frac{1}{2}\|h(x)\|^2$ weighted by the parameter $\omega > 0$ and $\|\cdot\|$ is the $\ell_2$-norm. The stochastic iterations, where noise is added at each iteration, will be introduced later. Condition (i) in Def. 2.1 guarantees that $\langle \nabla \mathcal{N}(x), \Psi(x) \rangle = 0$, so that the two terms in $\Lambda$ are orthogonal.

Note that we can use *any* relative descent directions as $\Psi$ depending on the specific problem. The Riemannian gradient in (7) is just one special case, which has some shortcomings. Firstly, it requires a potentially expensive projection $\mathrm{D}h(x)^* \left(\mathrm{D}h(x)^*\right)^\dagger$. Secondly, if the constraint involves a random noise on $h$, formula (7) it does not give an unbiased formula in expectation. An important contribution of the present work is the derivation of a computationally convenient form for the relative descent direction in the specific case of the generalized Stiefel manifold in Section 3.

We now turn to the analysis of the convergence of this method. The main object allowing for the convergence analysis is Fletcher's augmented Lagrangian

$$\mathcal{L}(x) = f(x) - \langle h(x), \lambda(x) \rangle + \beta \|h(x)\|^2, \tag{9}$$

with the Lagrange multiplier $\lambda(x) \in \mathbb{R}^p$ defined as $\lambda(x) = \left(\mathrm{D}h(x)^*\right)^\dagger [\nabla f(x)]$. The differential of $\lambda(x)$ must be smooth, which is met when $h$ is continuously differentiable and $\mathcal{M}^\varepsilon$ is a compact set.

**Assumption 2.3** (Multipliers of Fletcher's augmented Lagrangian). *The norm of the differential of the multipliers of Fletcher's augmented Lagrangian is bounded $\sup_{x \in \mathcal{M}^\varepsilon} \|\mathrm{D}\lambda(x)\| \leq C_\lambda$.*

**Proposition 2.3** (Lipschitz constant of Fletcher's augmented Lagrangian). *Fletcher's augmented Lagrangian $\mathcal{L}$ in (9) is $L_\mathcal{L}$-smooth on $\mathcal{M}^\varepsilon$, with $L_\mathcal{L} = L_{f+\lambda} + L_\mathcal{N}$, where $L_{f+\lambda}$ is the smoothness constant of $f(x) + \langle \lambda(x), h(x) \rangle$ and $L_\mathcal{N}$ is that of $\mathcal{N}(x)$.*

The following two lemmas show that there exists a positive step-size $\eta$, that guarantees that the next landing iteration stays within $\mathcal{M}^\varepsilon$ provided that the current iterate is inside of $\mathcal{M}^\varepsilon$.

**Lemma 2.4** (Upper bound on the safe step size). *Let $x \in \mathcal{M}^\varepsilon$ and consider the iterative update $\tilde{x} = x - \eta \Lambda(x)$, where $\eta > 0$ is a step size and $\Lambda(x)$ is the landing field with the weight parameter $\omega > 0$. If the step size satisfies*

$$\eta \leq \eta(x) := \frac{\omega \|\nabla \mathcal{N}(x)\|^2 + \sqrt{\omega^2 \|\nabla \mathcal{N}(x)\|^4 + L_\mathcal{N} \|\Lambda(x)\|^2 (\varepsilon^2 - \|h(x)\|^2)}}{L_\mathcal{N} \|\Lambda(x)\|^2}, \tag{10}$$

*where $L_\mathcal{N}$ is from Assumption 2.2, we have that the next iterate remains in the safe region $\tilde{x} \in \mathcal{M}^\varepsilon$.*

The proof can be found in the appendices in Subsection C.2. Next, we require that the norm of the relative descent direction must remain bounded in the safe region.

**Assumption 2.4** (Bounded relative descent direction). *We require that* $\sup_{x \in \mathcal{M}^\varepsilon} \|\Psi(x)\| \leq C_\Psi$.

This holds, for instance, if $\nabla f$ is bounded in $\mathcal{M}^\varepsilon$, using Def. 2.1 (ii) and Cauchy-Schwarz inequality. Under this assumption, we can lower bound the safe step in Lemma 2.4 for all $x \in \mathcal{M}^\varepsilon$, implying that there is always a positive step size that remains inside of the safe region.

**Lemma 2.5** (Non-disappearing safe step size). *The upper bound on the safe step size in Lemma 2.4 is lower bounded as* $\eta(x) \geq \min \left\{ \frac{\varepsilon}{\sqrt{2L_\mathcal{N}} C_\Psi}, \frac{\omega \bar{C}_h^2 \varepsilon^2}{L_\mathcal{N}(C_\Psi^2 + \omega^2 C_h \varepsilon^2)} \right\}$ *for all* $x \in \mathcal{M}^\varepsilon$ *where* $C_h, \bar{C}_h, C_\Psi > 0$ *are constants from Assumption 2.2 and 2.4.*

The proof can be found in Subsection C.3. The upper bound on the safe step size in Lemma 2.4 together with the statement of Lemma 2.5 that the upper bound remains positive for all $x \in \mathcal{M}^\varepsilon$, implying there is always a step size for the landing direction guaranteeing the iterates stay in $\mathcal{M}^\varepsilon$.

**Lemma 2.6.** *Let* $\mathcal{L}(x)$ *be Fletcher's augmented Lagrangian in* (9) *with* $\beta = (\frac{\rho}{4C_h^2} + \frac{C_\lambda}{2C_h} + \frac{C_\lambda^2}{C_h^2})/\omega$, *where* $\rho$ *is defined in Def. 2.1. We have that* $\langle \nabla \mathcal{L}(x), \Lambda(x) \rangle \geq \frac{\rho}{2} \left( \|\Psi(x)\|^2 + \|h(x)\|^2 \right)$.

The proof can be found in the appendices in Subsection C.4. This critical lemma shows that $\mathcal{L}$ is a Lyapunov function for the landing iterations and allows the study of convergence of the method with ease. The following statement combines Lemma 2.6 with the bound on the safe step size in Lemma 2.5 to prove sublinear convergence to a critical point on the manifold.

**Theorem 2.7** (Sublinear convergence). *The landing iteration in* (8) *starting from* $x_0 \in \mathcal{M}^\varepsilon$ *satisfies*

$$\frac{1}{K} \sum_{k=0}^{K} \|\Psi(x^k)\|^2 \leq 4 \frac{\mathcal{L}(x^0) - \mathcal{L}^*}{\eta \rho K} \qquad and \qquad \frac{1}{K} \sum_{k=0}^{K} \|h(x_k)\|^2 \leq 4 \frac{\mathcal{L}(x^0) - \mathcal{L}^*}{\eta \rho \omega^2 K}.$$

*for a fixed step size bounded as* $\eta \leq \min \left\{ \frac{\rho}{2L_\mathcal{L}}, \frac{\rho}{2L_\mathcal{L} C_h^2}, \frac{\varepsilon}{\sqrt{2L_\mathcal{N}} C_\Psi}, \frac{\omega \bar{C}_h^2 \varepsilon^2}{L_\mathcal{N}(C_\Psi^2 + \omega^2 C_h \varepsilon^2)} \right\}$.

Proof can be found in Subsection C.5. Due to the smoothness of Fletcher's augmented Lagrangian in the $\mathcal{M}^\varepsilon$ region, we can expand the convergence result to the stochastic setting, where the iterates are

$$x^{k+1} = x^k - \eta_k \left[ \Lambda(x^k) + \tilde{E}(x^k, \Xi^k) \right], \tag{11}$$

where the $\Xi^k$ are random i.i.d. variables and $\tilde{E}(x^k, \Xi^k)$ is the random error term at iteration $x^k$. As usual in stochastic optimization, we require that the error is unbiased and of bounded variance.

**Assumption 2.5** (Zero-centered and bounded variance). *There exists* $\gamma > 0$ *such that for all* $x \in \mathcal{M}^\varepsilon$, *we have* $\mathbb{E}_\Xi[\tilde{E}(x, \Xi)] = 0$ *and* $\mathbb{E}_\Xi[\|\tilde{E}(x, \Xi)\|^2] \leq \gamma^2$.

We obtain the following result with decaying step sizes.

**Theorem 2.8** (Stochastic landing). *Under Assumption 2.5, the landing iteration in* (11) *with step size* $\eta_k = \eta_0 \times (1 + k)^{-1/2}$ *produces iterates for which*

$$\inf_{k \leq K} \mathbb{E} \left[ \|\Psi(x_k)\|^2 \right] = \frac{4}{\rho \eta_0 \sqrt{K}} \left( \mathcal{L}(x^0) - \mathcal{L}^* + \frac{\eta_0 L_\mathcal{L} \gamma^2}{2} \log(K) \right)$$

$$\inf_{k \leq K} \mathbb{E} \left[ \|h(x)\|^2 \right] = \frac{4}{\rho \omega^2 \eta_0 \sqrt{K}} \left( \mathcal{L}(x^0) - \mathcal{L}^* + \frac{\eta_0 L_\mathcal{L} \gamma^2}{2} \log(K) \right),$$

*for the initial step size* $\eta_0 = \min \left\{ \frac{\rho}{2L_\mathcal{L}}, \frac{\rho}{2L_\mathcal{L} C_h^2}, \frac{\varepsilon}{\sqrt{2L_\mathcal{N}} C_\Psi}, \frac{\omega \bar{C}_h^2 \varepsilon^2}{L_\mathcal{N}(C_\Psi^2 + \omega^2 C_h \varepsilon^2)} \right\}$.

The theorem is proved in subsection C.6. We recover the same convergence rate as Riemannian SGD on the manifold in the non-convex setting.

# 3 LANDING ON THE GENERALIZED STIEFEL MANIFOLD

This section builds on the results of the previous Section 2 and proves that the simple landing update rule $X^{k+1} = X^k - \eta_k \Lambda(X^k)$, as defined as in (3), converges to the critical points of (1). The generalized Stiefel manifold $\text{St}_B(p, n)$ is defined by the constraint function $h(X) = X^\top B X - I_p$, and we have $\nabla \mathcal{N}(X) = 2BX(X^T BX - I_p)$. We now derive the quantities required for Assumption 2.2.

**Proposition 3.1** (Smoothness constants for the generalized Stiefel manifold). *The smoothness constants in Assumption 2.2 for the generalized Stiefel manifold are*

$$C_h = 2\sqrt{(1+\varepsilon)\kappa} \qquad and \qquad \bar{C}_h = 2\sqrt{(1-\varepsilon)\kappa^{-1}}, \qquad (12)$$

*where $\kappa$ is the condition number of $B$.*

Proof is presented in Subsection D.2. We show two candidates for the relative descent direction:

**Proposition 3.2** (Relative descent directions for the generalized Stiefel manifold). *The following three formulas are viable relative descent directions on the generalized Stiefel manifold.*

$$\Psi_B(X) = 2\text{skew}(\nabla f(X)X^\top B)BX \qquad (13)$$

$$\Psi_B^{\text{R}}(X) = 2\text{skew}(B^{-1}\nabla f(X)X^\top)BX \qquad (14)$$

*with $\Psi_B(X)$ having $\rho_B = 1/(\kappa(1+\varepsilon))$ and $\Psi_B^{\text{R}}(X)$ having $\rho_B^{\text{R}} = \beta_n^2/(\kappa(1+\varepsilon))$ for $\kappa = \beta_1/\beta_n$.*

Proof is given in Subsection D.3. The formula for the relative descent $\Psi_B^{\text{R}}(X)$ can be derived as a Riemannian gradient for $\text{St}_B(p, n)$ in a metric derived from a canonical metric on the standard Stiefel manifold via specific isometry; see Appendix E. The fact that $\Psi_B(X)$ above meets the conditions of Definition 2.1 allows us to define the deterministic landing iterations as $X^{k+1} = X^k - \eta^k \Lambda(X^k)$ with

$$\Lambda(X) = 2\,\text{skew}(\nabla f(X)X^\top B)BX + 2\omega BX(X^T BX - I_p) , \qquad (15)$$

and Theorem 2.7 applies to these iterations, showing that they converge to critical points.

## 3.1 STOCHASTIC GENERALIZED STIEFEL CASE

One of the main features of the formulation in (15) is that it seamlessly extends to the stochastic case when both the objective $f$ and the constraint matrix $B$ are expectations. Indeed, using the stochastic estimate $\Lambda_{\xi,\zeta,\zeta'}$ defined in Eq. (2), we have $\mathbb{E}_{\xi,\zeta,\zeta'}[\Lambda_{\xi,\zeta,\zeta'}(X)] = \Lambda(X)$. The stochastic landing iterations are, therefore, of the same form as section 2, (11). To apply Theorem 2.8 we need to bound the variance of $\tilde{E}(X, \Xi) = \Lambda_{\xi,\zeta,\zeta'}(X) - \Lambda(X)$ where the random variable $\Xi$ is the triplet $(\xi, \zeta, \zeta')$ using standard U-statistics techniques (Van der Vaart, 2000).

**Proposition 3.3** (Variance estimation of the generalized Stiefel landing iteration). *Let $\sigma_B^2$ be the variance of $B_\zeta$ and $\sigma_G^2$ the variance of $\nabla f_\xi(X)$. We have that*

$$\mathbb{E}_\Xi[\|\tilde{E}(X, \Xi)\|^2] \leq \sigma_G^2 p_B^2 \frac{(1+\varepsilon)^2}{\beta_n^2} + \sigma_B^2 \frac{1+\varepsilon}{\beta_n}\left(4\Delta(p_B + \beta_1^2) + p_N + (1+\varepsilon)^2\right), \qquad (16)$$

*with $p_B = \mathbb{E}[\|B_\zeta\|_2^2]$, $p_N = \frac{1+\varepsilon}{\beta_n}\sigma_B^2 + \varepsilon$ and $\Delta = \sup_{X \in \text{St}_B^\varepsilon(p,n)} \|\nabla f(X)X^\top\|_2^2$.*

The proof is found in subsection D.4. Note, that as expected, the variance in (16) cancels when both variances $\sigma_B$ and $\sigma_G$ cancel. A consequence of Proposition 3.3 is that Theorem 2.8 applies in the case of the stochastic landing method on the generalized Stiefel manifold.

**Proposition 3.4.** *For $L_\mathcal{L} = \mathcal{O}(L_f + L_\mathcal{N})$ we have that the asymptotic number of iterations the stochastic landing algorithm takes to achieve $e$-critical point for the generalized eigenvalue problem where $f(X) = -\frac{1}{2}\text{Tr}(X^\top AX)$ and $h(X) = X^\top BX - I$ is:*

$$\mathcal{O}\left(\left(\kappa\beta_1\sigma_G^2 + \left(1 + \beta_1^{-2}\right)\sigma_B^2\right)\beta_1\kappa^3(\kappa + \alpha_1 + \beta_1)\frac{np}{e^2}\right), \qquad (17)$$

*where $\alpha_i, \beta_i$ denote the eigenvalues of $A, B$ in decreasing order and $\kappa$ is the condition number of $B$.*

The proof is given in subsection D.5. Note that the bound above assumes $L_\mathcal{L} = \mathcal{O}(L_f + L_\mathcal{N})$, which are derived in Lemma D.1 and does not take into account the middle term of $\mathcal{L}(X)$ in (9).

## 4 NUMERICAL EXPERIMENTS

**Deterministic generalized eigenvalue problem.** We compare the methods on the top-$p$ generalized eigenvalue problem that consists of solving $\min_{X \in \mathbb{R}^{n \times p}} -\frac{1}{2}\text{Tr}(X^\top AX)$ for $X \in \text{St}_B(p, n)$. The two matrices are randomly generated with a condition number $\kappa = 100$ and with the size $n = 1000$ and $p = 500$. The matrix $A \in \mathbb{R}^{n \times n}$ is generated to have equidistant eigenvalues $\lambda(A)_i \in [1/\kappa, 1]$ and $B \in \mathbb{R}^{n \times n}$ has exponentially decaying eigenvalues $\lambda(B)_i \in [1/\kappa, 1]$.

Fig. 2 shows the timings of four methods with fixed stepsize: Riemannian steepest descent with QR-based Cholesky retraction (Sato & Aihara, 2019), the two landing methods with either $\Psi_B^R(X)$ and $\Psi_B(X)$ in Prop. 3.2, and the PLAM method (Gao et al., 2022a). We give the specifics of the experiment in Sec. B. All of the algorithms are implemented to be computed on a GPU using CUDA acceleration. The landing method with $\Psi_B(X)$ converges the fastest in terms of time, due to its cheap per-iteration computation, which is also demonstrated in Fig. 4 and Fig. 6 in the appendices. It can be also observed, that the landing method with $\Psi_B(X)$ is more robust to the choice of parameters $\eta$ and $\omega$ compared to PLAM, which we show in Fig. 7 and Fig. 9 in the appendices, and is in line with the equivalent observations for the orthogonal manifold (Ablin & Peyré, 2022, Fig. 9). Numerically tracking the value of the upper bound $\eta(X)$ of the safe stepsize from Lemma 2.4 shows that it is only mildly restricting at the start and becomes relaxed as the iterations approach a stationary point; see Fig. 8 in the appendices.

**Stochastic Canonical correlation analysis.** We use the standard benchmark problem for CCA, in which the MNIST dataset in split in half by taking left and right halves of each image, and compute the top-$p$ canonical correlation components by solving (4).

Fig. 3 shows the timings for the Riemannian gradient descent with rolling averaged covariance matrix and the landing algorithm with $\Psi_B(X)$ in its online and averaged form. The methods are implemented in PyTorch using CUDA. The averaged methods keep track of the covariance matrices during the first pass through the dataset, which is around 2.5 sec., after which they have the exact fully sampled covariance matrices. The online method has always only the sampled estimate with the batch size of $r = 512$. The stepsize is $\eta = 0.1$ and $\omega = 1$; in practice, the hyperparameters can be picked by grid-search as is common for stochastic optimization methods.

The online landing method outperforms the averaged Riemannian gradient descent in the online setting after only a few passes over the data, e.g. at the 2.5 sec. mark, which corresponds to the first epoch, at which point each sample appeared just once. After the first epoch, the rolling avg. methods get the advantage of the exact fully sampled covariance matrix and, consequently, have better distance $\mathcal{N}(X)$, but at the cost of requiring $\mathcal{O}(n^2)$ memory for the full covariance matrix. The online method does not form $B$ and requires only $\mathcal{O}(np)$ memory. The behavior is also consistent when $p = 10$ as shown in Fig. 5 in the appendices.

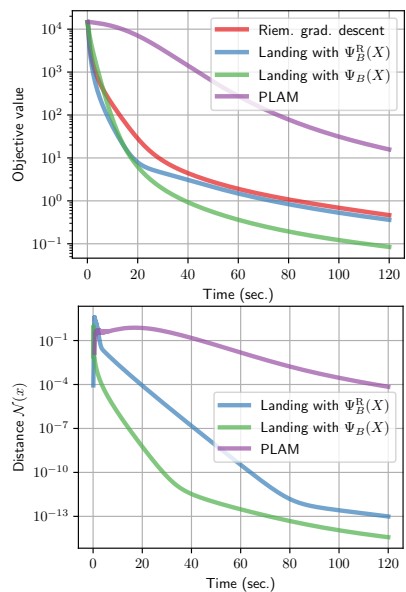

Figure 2: Deterministic computation of the generalized eigenvalue problem with $n = 1000, p = 500$, the condition number of the two matrices is $\kappa = 100$.

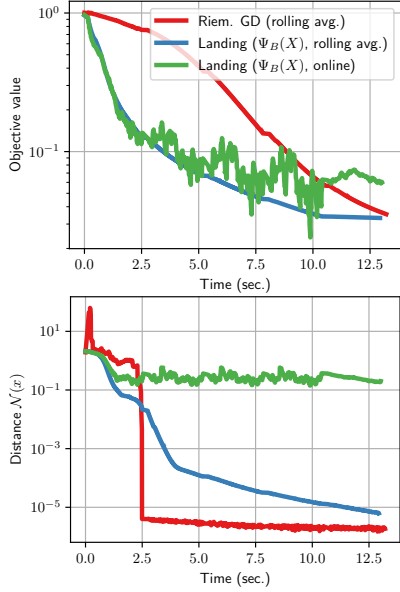

Figure 3: Stochastic CCA on the split MNIST dataset for $p = 5$. An epoch takes roughly 2.5 sec.

## 5 CONCLUSION

We extend the theory of the landing method from the Stiefel manifold to the general case of a smooth constraint $h(x) = 0$. We improve the existing analysis by using a smooth Lagrangian function, which allows us to also consider situations when we have only random estimates of the manifold, and we wish our iterates to be on the constraint in expectation. We show that random generalized Stiefel manifold, which is central to problems such as stochastic CCA and the GEVP, falls into the category of random manifold constraints and derive specific bounds for it. The analysis yields improved complexity bounds for stochastic CCA in a specific regime when the matrices are well-conditioned.

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
