## A SUMMARY OF RETRACTIONS ON THE GENERALIZED STIEFEL MANIFOLD

For an update to a matrix $X \in \mathrm{St}_B(p, n)$ following the direction $Z \in \mathbb{R}^{n \times p}$ there are several ways to compute a retraction.

- The *Polar decomposition* (Yger et al., 2012) uses

$$\mathrm{Retr}_{\mathrm{St}_B}(X, Z) = (X + Z)\left(I_p + Z^\top B Z\right)^{-1/2}, \tag{18}$$

where it is necessary to compute matrix product and a matrix square root inverse, amounting to $\mathcal{O}(n^2 p)$ flops.

- Mishra & Sepulchre (2016) observed that the aforementioned polar decomposition can be expressed as $UV^\top$ in terms of an SVD-like decomposition of $X + Z = U\Sigma V^\top$, where $U, V$ are orthogonal in respect to $B$-inner product, whose main cost is the eigendecomposition of $(X + Z)^\top B (X + Z)$.

- Recently, Sato & Aihara (2019) proposed the *Cholesky-QR based retraction*

$$\mathrm{Retr}_{\mathrm{St}_B}(X, Z) = (X + Z)R^{-1}, \tag{19}$$

where $R \in \mathbb{R}^{p \times p}$ comes from Cholesky factorization of $R^\top R = (X + Z)^\top B (X + Z)$. The flops required for the computation amount to $\mathcal{O}(n^2 p)$, which comes from the matrix multiplications, the Cholesky factorization of an $p \times p$ matrix, and finally, the inverse multiplication by a small triangular $p \times p$ matrix requires $\mathcal{O}(p^3)$ to form and $\mathcal{O}(np^2)$ to multiply with.

## B ADDITIONAL EXPERIMENTS AND FIGURES

For the experiment showed in Fig. 2, we pick the step-size $\eta$ parameter to be $\eta = 0.01$ for the Riemannian gradient descent, the landing with $\Psi_B^R(X)$, and PLAM, and $\eta = 200$ for the landing with $\Psi_B(X)$. The normalizing parameter $\omega$ is chosen to be $\omega = 10^5$ for the landing with $\Psi_B^R(X)$, $\omega = 0.1$ for the landing with $\Psi_B(X)$, and $\omega = 200$ for PLAM.

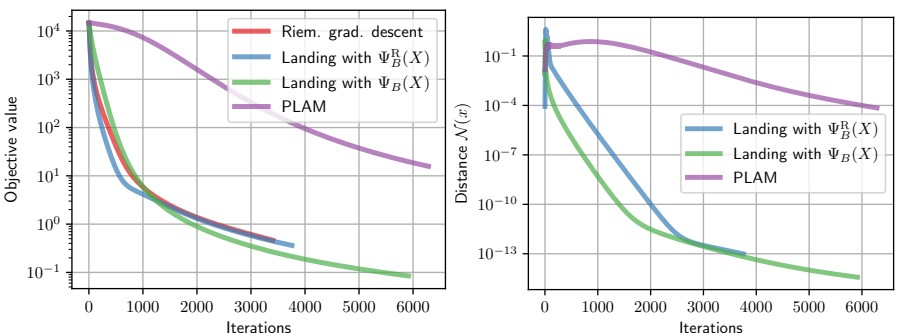

Figure 4: Deterministic computation of the generalized eigenvalue problem with $n = 1000, p = 500$, the condition number of the two matrices $\kappa = 100$. Each algorithm is given a time limit of 120 seconds.

## C PROOFS FOR SECTION 2

### C.1 PROOF OF PROPOSITION 2.2

*Proof.* It follows from the definition (7) and $\mathrm{D}h(x)\mathrm{D}h(x)^* (\mathrm{D}h(x)^*)^\dagger = \mathrm{D}h(x)$ that $\mathrm{D}h(x)(\mathrm{grad}f(x)) = 0$, which implies the first condition in Definition 2.1 holds, i.e., $\langle \mathrm{grad}f(x), v \rangle = 0$ for all $v \in \mathrm{span}(\mathrm{D}h(x)^*)$. Since $\mathrm{D}h(x)^* (\mathrm{D}h(x)^*)^\dagger \nabla f(x) \in \mathrm{span}(\mathrm{D}h(x)^*)$,

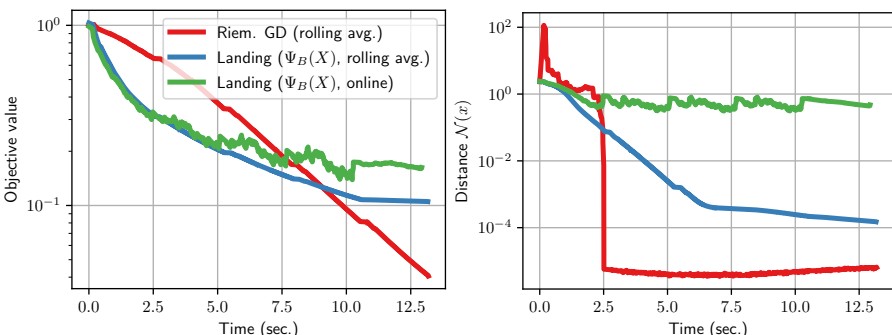

Figure 5: Stochastic canonical correlation analysis on the split MNIST dataset for $p = 5$ canonical components.

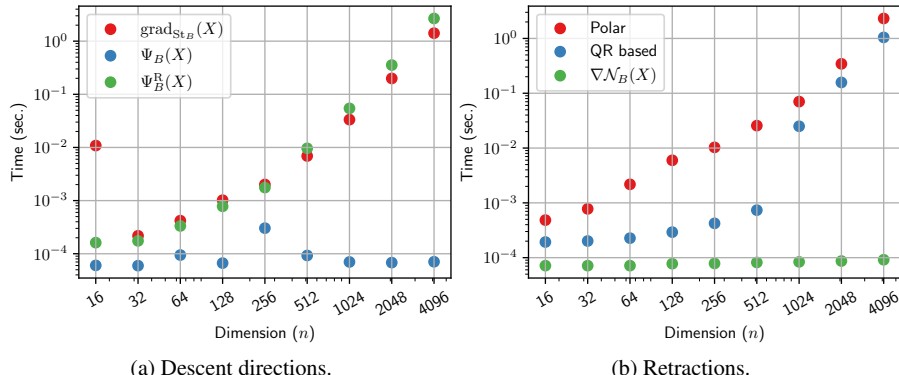

(a) Descent directions.

(b) Retractions.

Figure 6: Comparison of per-iteration computational time for different problem sizes of the descent directions of algorithms in Fig. 2 and the cost of retractions compared to $\nabla\mathcal{N}(X)$, both in the deterministic setting when $n = p = r$, for which the matrix multiplication in $\Psi_B(X)$ and $\nabla_{\mathcal{N}}(X)$ are at the disadvantage. Computation time of randomly generated $B, X \in \mathbb{R}^{n \times n}$ averaged over 100 runs with CUDA implementation using cupy.

we have

$$
\begin{aligned}
\|\mathrm{grad} f(x)\|^2 &= \langle \mathrm{grad} f(x), \mathrm{grad} f(x) \rangle \\
&= \left\langle \mathrm{grad} f(x), \nabla f(x) - \mathrm{D}h(x)^* \left(\mathrm{D}h(x)^*\right)^\dagger \nabla f(x) \right\rangle \\
&= \langle \mathrm{grad} f(x), \nabla f(x) \rangle,
\end{aligned}
$$

which verifies the second condition with $\rho = 1$ and the third condition with $\mathrm{grad} f(x) = 0$ for a critical point $x \in \mathcal{M}$.  $\square$

## C.2  PROOF OF LEMMA 2.4

*Proof.* For ease of notation we denote the current iterate $x$ and the subsequent iterate as $\tilde{x} = x - \eta\Lambda(x)$. From $L_{\mathcal{N}}$-Lipschitz of $\mathcal{N}$ we have

$$
\mathcal{N}(\tilde{x}) \leq \mathcal{N}(x) + \langle \nabla\mathcal{N}(x), -\eta\Lambda(x)\rangle + \frac{\eta^2 L_{\mathcal{N}}}{2}\|\Lambda(x)\|^2 \tag{20}
$$

$$
= \mathcal{N}(x) - \eta\omega\|\nabla\mathcal{N}(x)\|^2 + \frac{\eta^2 L_{\mathcal{N}}}{2}\|\Lambda(x)\|^2, \tag{21}
$$

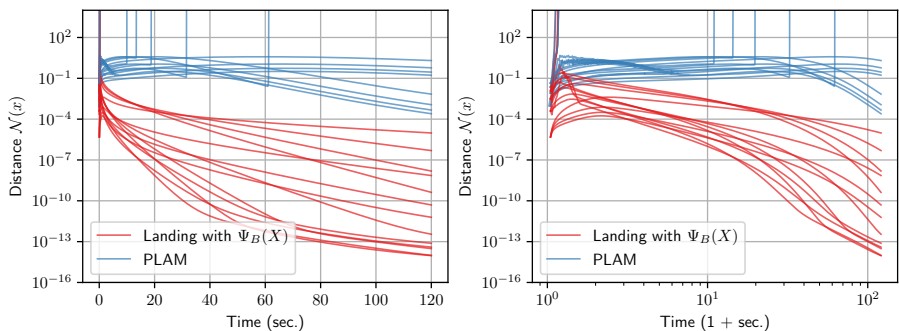

Figure 7: Comparison of the sensitivity to the choice of the step-size $\eta$ and $\omega$ of the landing with $\Psi_B(X)$ and the PLAM method (Gao et al., 2022a) in the generalized eigenvalue problem experiment presented in Fig. 2 with $n = 1000, p = 500$, and the condition number of the two matrices $\kappa = 100$. On the right we show log-log scale to better see the effect in earlier iterations. Both parameters are picked as in the experiment for Fig. 2 and multiplied by a scalar from the set $\{0.25, 0.75, 1.25, 1.75\}$ for all possible pair combinations.

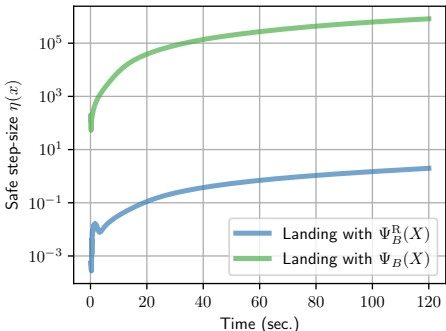

Figure 8: Numerical evaluation of the upper safe-step bound $\eta(X)$ in Lemmma 2.4 per time, which ensures that the iterates stay in $\mathrm{St}_B^\varepsilon(p, n)$, for the two landing methods tested in Fig. 2 with the $L_\mathcal{N}$ bounded for the GEVP as in Lemma D.1. We see that the upper bound is only mildly restricting and becomes even less restricting as the iterates come close to a stationary point.

where in the first line we use that $\mathcal{N}(x)$ has Lipschitz gradient with the constants $L_\mathcal{N}$ for $x$ in the safe-region. To guarantee $h(\tilde{x}) \leq \varepsilon$, we have to ensure that

$$\mathcal{N}(x) - \eta\omega\|\nabla\mathcal{N}(x)\|^2 + \frac{\eta^2 L_\mathcal{N}}{2}\|\Lambda(x)\|^2 \leq \frac{\varepsilon^2}{2}. \tag{22}$$

Solving the quadratic inequality in (22) for the positive root $\eta > 0$ yields the results. □

### C.3 PROOF OF LEMMA 2.5

*Proof.* Assume that $\|\nabla\mathcal{N}(x)\| \geq \bar{C}_h\|h(x)\|$ is lower bounded in $\mathcal{M}^\varepsilon$. We proceed to lower bound the numerator of the safe-step size bound in Lemma 2.4 by making it independent of $x \in \mathcal{M}^\varepsilon$ as follows

$$\omega\|\nabla\mathcal{N}(x)\|^2 + \sqrt{\omega^2\|\nabla\mathcal{N}(x)\|^4 + L_\mathcal{N}\|\Lambda(x)\|^2(\varepsilon^2 - \|h(x)\|^2)}$$

$$\geq \omega\bar{C}_h^2\|h(x)\|^2 + \sqrt{\omega^2\bar{C}_h^4\|h(x)\|^4 + L_\mathcal{N}\|\Psi(x)\|^2(\varepsilon^2 - \|h(x)\|^2)} \tag{23}$$

$$\geq \omega\bar{C}_h^2\|h(x)\|^2\left(1 + \frac{1}{\sqrt{2}}\right) + \frac{1}{\sqrt{2}}\|\Psi(x)\|\sqrt{L_\mathcal{N}(\varepsilon^2 - \|h(x)\|^2)} \tag{24}$$

$$\geq \sqrt{\frac{L_\mathcal{N}}{2}}\|\Psi(x)\|(\varepsilon - \|h(x)\|) + \left(1 + \frac{1}{\sqrt{2}}\right)\omega\bar{C}_h^2\|h(x)\|^2 \tag{25}$$

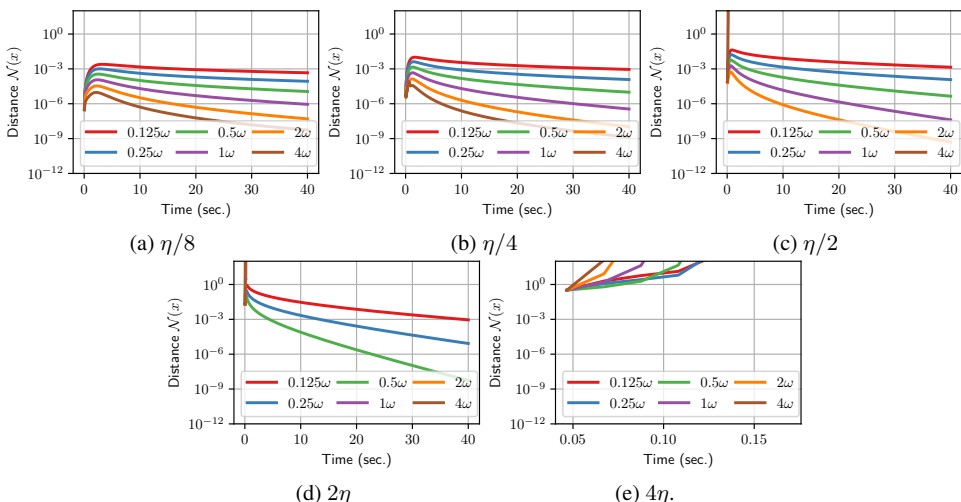

Figure 9: Robustness of the convergence towards the $\mathrm{St}_B(p, n)$ for the landing with $\Psi_B(X)$ in the experiment for Fig. 2 based on the multiplied perturbations of $\eta$ and $\omega$ parameters with the values from $\{1/8, 1/4, 1/2, 2, 4\}$.

where the first inequality comes from using bounds from Assumption 2.2, the second inequality comes from $\sqrt{a+b} \geq (\sqrt{a} + \sqrt{b})/\sqrt{2}$ for $a, b \geq 0$, and the final inequality from the fact that $\sqrt{a-b} \geq \sqrt{a} - \sqrt{b}$ for $a, b \geq 0$ and $a \geq b$. As a result we have that the upper bound in Lemma 2.4 is lower-bounded by

$$\eta(x) \geq \frac{\sqrt{\frac{L_{\mathcal{N}}}{2}} \|\Psi(x)\|(\varepsilon - \|h(x)\|) + \left(1 + \frac{1}{\sqrt{2}}\right) \omega \bar{C}_h^2 \|h(x)\|^2}{L_{\mathcal{N}} \left(\|\Psi(x)\|^2 + \omega^2 C_h^2 \|h(x)\|^2\right)}, \tag{26}$$

using the fact that $\|\Lambda(x)\|^2 = \|\Psi(x)\|^2 + \omega^2 \|\nabla\mathcal{N}(x)\|^2$ and $\|\nabla\mathcal{N}(x)\|^2 \leq C_h^2 \|h(x)\|^2$. Since the minimum of (26) in terms of $\|h(x)\| \in [0, \varepsilon]$ is on the boundary, when $\|h(x)\| = 0$ or $\|h(x)\| = \varepsilon$, we can further lower bound the safe step size as

$$\eta(x) \geq \min\left\{\frac{\varepsilon}{\sqrt{2L_{\mathcal{N}}}C_\Psi}, \frac{\omega \bar{C}_h^2 \varepsilon^2}{L_{\mathcal{N}}(C_\Psi^2 + \omega^2 C_h \varepsilon^2)}\right\} \tag{27}$$

where we used for the minimum at $\|h(x)\| = 0$ and the bound $\sup_{x \in \mathcal{M}^\varepsilon} \|\Psi(x)\| \leq C_\Psi$. $\qquad\square$

## C.4 PROOF OF LEMMA 2.6

*Proof.* The inner product has two parts

$$\langle \nabla\mathcal{L}(x), \Lambda(x) \rangle = \mathrm{D}\mathcal{L}(x)[\Lambda(x)] = \mathrm{D}\mathcal{L}(x)[\Psi(x)] + \omega\mathrm{D}\mathcal{L}(x)[\nabla\mathcal{N}(x)]. \tag{28}$$

We expand the first term in (28) as

$$\mathrm{D}\mathcal{L}(x)[\Psi(x)] = \langle \nabla f(x), \Psi(x) \rangle - \left\langle (\mathrm{D}h(x)^*)^\dagger \nabla f(x), \mathrm{D}h(x)\Psi(x) \right\rangle \tag{29}$$

$$- \langle \mathrm{D}\lambda(x)[\Psi(x)], h(x) \rangle + 2\beta \langle \nabla\mathcal{N}(x), \Psi(x) \rangle \tag{30}$$

$$= \langle \nabla f(x), \Psi(x) \rangle - \langle \mathrm{D}\lambda(x)[\Psi(x)], h(x) \rangle \tag{31}$$

where we use that $\nabla\|h(x)\|^2 = 2\nabla\mathcal{N}(x)$ and that the second and the third term are zero due to the orthogonality of $\Psi(x)$ with the span of $\mathrm{D}h(x)^*$. We expand the second term in (28) as

$$\mathrm{D}\mathcal{L}(x)[\nabla\mathcal{N}(x)] = \langle \nabla f(x), \nabla\mathcal{N}(x) \rangle - \left\langle (\mathrm{D}h(x)^*)^\dagger \nabla f(x), \mathrm{D}h(x)\nabla\mathcal{N}(x) \right\rangle \tag{32}$$

$$- \langle \mathrm{D}\lambda(x)[\nabla\mathcal{N}(x)], h(x) \rangle + 2\beta\|\nabla\mathcal{N}(x)\|^2 \tag{33}$$

$$= \left\langle (I_n - \mathrm{D}h(x)^*(\mathrm{D}h(x)^*)^\dagger)\nabla f(x), \nabla\mathcal{N}(x) \right\rangle \tag{34}$$

$$- \langle \mathrm{D}\lambda(x)[\nabla\mathcal{N}(x)], h(x) \rangle + 2\beta\|\nabla\mathcal{N}(x)\|^2 \tag{35}$$

$$= - \langle \mathrm{D}\lambda(x)[\nabla\mathcal{N}(x)], h(x) \rangle + 2\beta\|\nabla\mathcal{N}(x)\|^2 \tag{36}$$

where in the second equality we move the adjoint $\mathrm{D}h(x)^*$ in the second inner product to the left side and join it with the first inner product. The third equality comes from the fact that due to the projection of $\nabla f(x)$ on the orthogonal complement of $\mathrm{D}h(x)^*$ and $\nabla \mathcal{N}(x) = \mathrm{D}h(x)^* h(x)$ are orthogonal.

Joining the two components (31) and (36) together we get

$$\langle \nabla \mathcal{L}(x), \Lambda(x) \rangle = \langle \nabla f(x), \Psi(x) \rangle - \langle \mathrm{D}\lambda(x)[\Lambda(x)], h(x) \rangle + 2\beta\omega \|\nabla\mathcal{N}(x)\|^2 \tag{37}$$

$$\geq \rho\|\Psi(x)\|^2 - C_\lambda(\|\Psi(x)\| + \omega\|\nabla\mathcal{N}(x)\|)\|h(x)\| + 2\beta\omega\|\nabla\mathcal{N}(x)\|^2 \tag{38}$$

$$\geq \rho\|\Psi(x)\|^2 + \omega(2\beta C_h - C_\lambda)C_h\|h(x)\|^2 - C_\lambda\|\Psi(x)\|\|h(x)\| \tag{39}$$

$$\geq \rho\|\Psi(x)\|^2 + \omega(2\beta C_h - C_\lambda)C_h\|h(x)\|^2 - C_\lambda\left(\alpha\|\Psi(x)\|^2 + \alpha^{-1}\|h(x)\|^2\right) \tag{40}$$

$$\geq (\rho - C_\lambda\alpha)\|\Psi(x)\|^2 + (2\omega\beta C_h^2 - \omega C_h C_\lambda - \alpha^{-1}C_\lambda)\|h(x)\|^2 \tag{41}$$

$$\geq \frac{\rho}{2}\left(\|\Psi(x)\|^2 + \|h(x)\|^2\right) \tag{42}$$

where the first inequality comes from $\langle \nabla f(x), \Psi(x) \rangle \geq \rho\|\Psi(x)\|^2$ in Def. 2.1 combined with the bound $\sup_{x \in \mathcal{M}^\varepsilon} \|\mathrm{D}\lambda(x)\| \leq C_\lambda$ and the triangle inequality, the second inequality comes from bounding $\|\nabla\mathcal{N}(x)\| \leq C_h\|h(x)\|$ using Assumption 2.2 and rearranging terms, the third inequality comes from using the AG-inequality $\sqrt{ab} \leq (a + b)/2$ with $a = \alpha\|h(x)\|$ and $b = \alpha^{-1}\|\Psi(x)\|$ for an arbitrary $\alpha > 0$, in the fourth inequality we only rearrange terms, and finally, in the fifth inequality we choose $\alpha = \rho/(2C_\lambda)$ and use that $\beta = \left(\frac{\rho}{4C_h^2} + \frac{C_\lambda}{2C_h} + \frac{C_\lambda^2}{C_h^2}\right)/\omega$. □

## C.5   Proof of Theorem 2.7

*Proof.* Due to $x_0 \in \mathcal{M}^\varepsilon$ and the step size being smaller than the bound in Lemma 2.5, we have that all iterates remain in the safe region $x^k \in \mathcal{M}^\varepsilon$. By smoothness of Fletcher's augmented Lagrangian we can expand

$$\mathcal{L}(x^{k+1}) \leq \mathcal{L}(x^k) - \eta\langle \Lambda(x^k), \nabla\mathcal{L}(x^k) \rangle + \frac{L_\mathcal{L}\eta^2}{2}\|\Lambda(x^k)\|^2 \tag{43}$$

$$\leq \mathcal{L}(x^k) - \frac{\eta\rho}{2}\left(\|\Psi(x^k)\|^2 + \omega^2\|h(x^k)\|^2\right) + \frac{L_\mathcal{L}\eta^2}{2}\|\Lambda(x^k)\|^2 \tag{44}$$

$$\leq \mathcal{L}(x^k) - \frac{\eta}{2}\left((\rho - L_\mathcal{L}\eta)\|\Psi(x^k)\|^2 + \omega^2\left(\rho - \eta L_\mathcal{L}C_h^2\right)\|h(x^k)\|^2\right), \tag{45}$$

where in the second inequality we used the results of Lemma 2.6, and in the third inequality we use the bound on $\|\nabla\mathcal{N}(x)\| \leq C_h\|h(x)\|$ by Assumption 2.2. By the step size $\eta < \min\left\{\frac{\rho}{2L_\mathcal{L}}, \frac{\rho}{2L_\mathcal{L}C_h^2}\right\}$ we have

$$\frac{\eta\rho}{4}\|\Psi(x^k)\|^2 + \frac{\eta\rho\omega^2}{4}\|h(x)\|^2 \leq \mathcal{L}(x^k) - \mathcal{L}(x^{k+1}). \tag{46}$$

Telescopically summing the first $K$ terms gives

$$\frac{\eta\rho}{4}\sum_{k=0}^{K}\|\Psi(x^k)\|^2 + \frac{\eta\rho\omega^2}{4}\sum_{k=0}^{K}\|h(x)\|^2 \leq \mathcal{L}(x^0) - \mathcal{L}(x^{K+1}) \leq \mathcal{L}(x^0) - \mathcal{L}^*, \tag{47}$$

which implies that the inequalities hold individually also

$$\frac{\eta\rho}{4}\sum_{k=0}^{K}\|\Psi(x^k)\|^2 \leq \mathcal{L}(x^0) - \mathcal{L}^* \quad \text{and} \quad \frac{\eta\rho\omega^2}{4}\sum_{k=0}^{K}\|h(x)\|^2 \leq \mathcal{L}(x^0) - \mathcal{L}^*. \tag{48}$$

□

## C.6 Proof of Theorem 2.8

*Proof.* By the Lipschitz continuity of the gradient of Fletcher's augmented Lagrangian we have

$$\mathbb{E}\left[\tilde{\mathcal{L}}(x^{k+1})\right] \leq \mathbb{E}\left[\mathcal{L}(x^k) - \eta\left\langle\tilde{\Lambda}(x^k), \nabla\mathcal{L}(x^k)\right\rangle + \frac{L_{\mathcal{L}}\eta^2}{2}\|\tilde{\Lambda}(x^k)\|^2\right] \tag{49}$$

$$\leq \mathcal{L}(x^k) - \eta\left\langle\Lambda(x^k), \nabla\mathcal{L}(x^k)\right\rangle + \frac{L_{\mathcal{L}}\eta^2}{2}\left(\|\Lambda(x^k)\|^2 + \gamma^2\right) \tag{50}$$

$$\leq \mathcal{L}(x^k) - \frac{\eta\rho}{2}\left(\|\Psi(x^k)\|^2 + \omega^2\|h(x)\|^2\right) + \frac{L_{\mathcal{L}}\eta^2}{2}\left(\|\Lambda(x^k)\|^2 + \gamma^2\right) \tag{51}$$

$$\leq \mathcal{L}(x^k) - \frac{\eta}{2}\left((\rho - L_{\mathcal{L}}\eta)\|\Psi(x^k)\|^2 + \omega^2\left(\rho - \eta L_{\mathcal{L}}C_h^2\right)\|h(x^k)\|^2\right) + \frac{L_{\mathcal{L}}\eta^2}{2}\gamma^2, \tag{52}$$

where the first inequality comes from taking an expectation of the Lipschitz-continuity of $\mathcal{L}(x)$, in the second inequality we take the expectation inside of the inner product using the fact that $\tilde{\Lambda}(x^k)$ is zero-centered and has bounded variance, the third and the last inequality comes as a consequence of Lemma 2.6. By the step size being smaller than $\eta < \min\left\{\frac{\rho}{2L_{\mathcal{L}}}, \frac{\rho}{2L_{\mathcal{L}}C_h^2}\right\}$

$$\frac{\eta\rho}{4}\|\Psi(x^k)\|^2 + \frac{\eta\rho\omega^2}{4}\|h(x)\|^2 \leq \mathcal{L}(x^k) - \mathcal{L}(x^{k+1}) + \frac{L_{\mathcal{L}}\eta^2}{2}\gamma^2 \tag{53}$$

Telescopically summing the first $K$ terms gives

$$\frac{\eta\rho}{4}\sum_{k=1}^{K}\|\Psi(x^k)\|^2 + \frac{\eta\rho\omega^2}{4}\sum_{k=0}^{K}\|h(x^k)\|^2 \leq \mathcal{L}(x^0) - \mathcal{L}(x^{K+1}) + \frac{L_{\mathcal{L}}\eta^2\gamma^2}{2}\sum_{k=0}^{K}(1+k)^{-1} \tag{54}$$

$$\leq \mathcal{L}(x^0) - \mathcal{L}^* + \frac{L_{\mathcal{L}}\eta^2\gamma^2}{2}\log(K) \tag{55}$$

which implies that the inequalities hold also individually

$$\inf_{k\leq K}\|\Psi(x^k)\|^2 \leq \frac{4}{\rho\eta_0\sqrt{K}}\left(\mathcal{L}(x^0) - \mathcal{L}^* + \frac{\eta_0 L_{\mathcal{L}}\gamma^2}{2}\log(K)\right) \tag{56}$$

$$\inf_{k\leq K}\|h(x^k)\|^2 \leq \frac{4}{\rho\omega^2\eta_0\sqrt{K}}\left(\mathcal{L}(x^0) - \mathcal{L}^* + \frac{\eta_0 L_{\mathcal{L}}\gamma^2}{2}\log(K)\right), \tag{57}$$

where we used that $\inf_{k\leq K}\|\Psi(x^k)\|^2 \leq \sum_{k=0}^{K}\eta_k\|\Psi(x^k)\|^2\left(\sum_{k=0}^{K}\eta_k\right)^{-1}$ and the fact that $\sum_{k\leq K}\eta_k \geq \eta_0\sqrt{K}$. $\square$

## D Proofs for Section 3

### D.1 Specific forms of $\mathrm{D}h(x), \lambda(X)$ for $\mathrm{St}_B(p,n)$

We begin by showing the specific form of the formulations derived in the previous section for the case of the generalized Stiefel manifold. Differentiating the generalized Stiefel constraint yields $\mathrm{D}h(X)[V] = X^\top BV + V^\top BX$ and the adjoint is derived as

$$\left\langle\mathrm{D}h(X)^*[V], W\right\rangle = \left\langle V, \mathrm{D}h(X)[W]\right\rangle = \left\langle V, W^TBX + X^TBW\right\rangle = \left\langle 2BX\mathrm{sym}(V), W\right\rangle, \tag{58}$$

as such we have that $\mathrm{D}h(X)^*[V] = 2BX\mathrm{sym}(V)$. Consequently

$$\mathrm{D}h(X)\mathrm{D}h(X)^*[V] = 2\mathrm{sym}(V)X^\top B^2X + 2X^\top B^2X\mathrm{sym}(V), \tag{59}$$

and the Lagrange multiplier $\lambda(X)$ is defined in the case of the generalized Stiefel manifold as the solution to the following Lyapunov equation

$$\lambda(X)X^\top B^2X + X^\top B^2X\lambda(X) = X^\top B\nabla f(X) + \nabla f(X)^\top BX. \tag{60}$$

Importantly, due to $\lambda(X)$ being the unique solution to the linear equation and $\nabla f(X)$ being smooth, $\lambda(X)$ is also smooth with respect to $X$, and as a smooth function defined over a compact set $\mathrm{St}_B^\varepsilon(p,n)$, its operator norm is bounded $\sup_{X\in\mathrm{St}_B^\varepsilon(p,n)}\|\mathrm{D}\lambda(X)\|_F \leq C_\lambda$ as required by Assumption 2.3.

## D.2 Proof of Proposition 3.1

*Proof.* For $\|X^\top BX - I_p\|_F \leq \varepsilon$, $X = U\Sigma V^\top$ be the singular value decomposition of $X$, and $QDQ^\top$ be the spectral decomposition of $B$. We then have

$$\varepsilon^2 \geq \|X^\top BX - I_p\|_F^2 = \|\Sigma U^\top QD(U^\top Q)^\top \Sigma - I_p\|_F^2 \tag{61}$$

where $\beta_i, \sigma_i$ are the positive eigenvalues of $B$ and the singular values of $X$ respectively in the decreasing order. This implies that

$$\sqrt{(1-\varepsilon)/\beta_1} \leq \sigma_i \leq \sqrt{(1+\varepsilon)/\beta_n}. \tag{62}$$

The above bound gives that the singular values of $Dh(X)^* = 2BX$ are in the interval $[2\sqrt{(1-\varepsilon)\kappa^{-1}}, 2\sqrt{(1+\varepsilon)\kappa}]$ which in turn gives the constants $C_h, \bar{C}_h$. $\qquad\square$

**Lemma D.1** (Lipschitz constants for the generalized eigenvalue problem). *Let $f = -\frac{1}{2}\text{Tr}(X^\top AX)$ and $\mathcal{N}(X) = \|X^\top BX - I_p\|_F^2$ as in the optimization problem corresponding to the generalized eigenvalue problem. We have that, for $X \in \text{St}_B^\varepsilon(p, n)$, the Lipschitz constant for $\nabla \mathcal{N}$ is $L_\mathcal{N} = \beta_1 \varepsilon + 2(1+\varepsilon)\kappa$ and the for $\nabla f$ is $L_f = \alpha_1$ where $\alpha_1$ is the largest eigenvalue of $A$.*

*Proof.* Take $X, Y \in \text{St}_B(p, n)$, we have that $\nabla \mathcal{N}(X) = BX(X^\top BX)$, thus

$$\nabla \mathcal{N}(X) - \nabla \mathcal{N}(Y) = B\left(X(X^\top BX - I_p) - Y(Y^\top BY)\right) \tag{63}$$

$$= B(X-Y)(X^\top BX - I_p) + B(X^\top BX - Y^\top BY) \tag{64}$$

$$= B(X-Y)(X^\top BX - I_p) + BY(X-Y)^\top BX + BYY^\top B(X-Y) \tag{65}$$

Taking the Frobenius norm and by the triangle inequality we get

$$\|\nabla \mathcal{N}(X) - \nabla \mathcal{N}(Y)\| \leq \|X-Y\|\left(\|B\|\|X^\top BX - I_p\| + \|B\|^2\|X\|\|Y\| + \|B\|^2\|Y\|^2\right) \tag{66}$$

$$\leq \|X-Y\|(\beta_1 \varepsilon + 2(1+\varepsilon)\kappa), \tag{67}$$

where we used the fact that $X \in \text{St}_B^\varepsilon(p, n)$ and we have that $\|X\|_2 \leq \sqrt{(1+\varepsilon)\kappa}$ and the same for $Y \in \text{St}_B^\varepsilon(p, n)$.

When $f = \frac{1}{2}\text{Tr}(X^\top AX)$, we have that $\|\nabla f(X) - \nabla f(Y)\| \leq \|A\|_2\|X-Y\|$.

$\qquad\square$

## D.3 Proof of Proposition 3.2

*Proof.* For ease of notation we denote $G = \nabla f(X) \in \mathbb{R}^{n \times p}$. The first property Definition 2.1 *(i)* comes from

$$\langle \text{skew}(GX^\top B)BX, BXS \rangle = 0, \tag{68}$$

which holds for a symmetric matrix $S$, since a skew-symmetric matrix is orthogonal in the trace inner product to a symmetric matrix,

The second property *(ii)* is a consequence of the following

$$\langle \Psi_B(X), G \rangle = \langle \text{skew}(GX^TB)BX, G \rangle = \|\text{skew}(GX^TB)\|_F^2 \geq \frac{1}{(1+\varepsilon)\kappa}\|\Psi_B(X)\|_F^2, \tag{69}$$

where we use the bounds on $\|BX\|_2 \leq \sqrt{(1+\varepsilon)\kappa}$ derived in the proof of Proposition 3.1 in (62) for $\kappa = \beta_1/\beta_n$ the condition number of $B$.

To show the third property *(iii)*, we first consider a critical point $X \in \text{St}_B(p, n)$, for which must hold

$$G = BXS, \tag{70}$$

for some $S \in \text{sym}(p)$ due to the constraints being symmetric and that $X^\top BX = I_p$ by feasibility. We have that at the critical point defined in (70), the relative descent direction is

$$\Psi_B(X) = \text{skew}(GX^\top B)BX = \text{skew}(BXSX^\top B)BX = 0, \tag{71}$$

where the second equality is the consequence of (70) and the third equality comes from the fact that $BXSX^\top B$ is symmetric.

To show the other side of the implication, that $\Psi_B(X) = 0$ combined with feasibility imply that $X$ is a critical point, we consider

$$0 = \Psi(x) = \text{skew}(GX^\top B)BX = GX^\top B^2 X - BXG^\top BX \tag{72}$$

which, since $X^\top B^2 X \in \mathbb{R}^{p \times p}$ is invertible, is equivalent to

$$G = BXG^\top BX \left( X^\top B^2 X \right)^{-1}. \tag{73}$$

For $X$ to be a critical point, we must have that $G^\top BX \left( X^\top B^2 X \right)^{-1}$ is symmetric:

$$G^\top BX \left( X^\top B^2 X \right)^{-1} = \left( X^\top B^2 X \right)^{-1} X^\top BG, \tag{74}$$

which, after multiplying by $\left( X^\top B^2 X \right)$ from both sides and rearranging terms, is equivalent to

$$X^\top B\text{skew}(BXG^\top)BX = 0, \tag{75}$$

that is true from multiplying (72) by $X^\top B$ from the left.

For the other choice of relative gradient $\Psi_B^\text{R}(X) = \text{skew}(B^{-1}GX^\top)BX$, letting $M = B^{-1}GX^\top$, we find

$$\langle \Psi_B^\text{R}(X), G \rangle = \langle \text{skew}(M), BMB \rangle \tag{76}$$
$$= \langle \text{skew}(M), \text{skew}(BMB) \rangle \tag{77}$$
$$= \langle \text{skew}(M), B\text{skew}(M)B \rangle \tag{78}$$
$$\geq \|\text{skew}(M)\|_\text{F}^2 \beta_n^2 \tag{79}$$

and similarly as before, it holds $\|\Psi_B^\text{R}(X)\|^2 \leq \|\text{skew}(M)\|_\text{F}^2(1 + \varepsilon)\kappa$ which in turn leads to $\langle \Psi_B^\text{R}(X), G \rangle \geq \frac{\beta_n^2}{(1+\varepsilon)\kappa}\|\Psi_B\|^2$ $\qquad\qquad\qquad\qquad\qquad\qquad\qquad\qquad \square$

### D.4 PROOF OF PROPOSITION 3.3

*Proof.* We start by deriving the bound on the variance of the normalizing component $\nabla\mathcal{N}(X)$. Consider $U$ and $V$ to be two independent random matrices taking i.i.d. values from the distribution of $B_\zeta$ with variance $\sigma_B^2$. We have that

$$\text{Var}\left[ UX(X^\top VX - I_p) \right] = \mathbb{E}_{U,V}\left[ \|UX(X^\top VX - I_p) - BX(X^\top BX - I_p)\|^2 \right] \tag{80}$$

Introducing the random marginal $BX(X^\top VX - I_p)$, we further decompose

$$\text{Var}\left[ UX(X^\top VX - I_p) \right] = \mathbb{E}_{U,V}\left[ \|UX(X^\top VX - I_p) - BX(X^\top VX - I_p)\|^2 \right] \tag{81}$$
$$+ \mathbb{E}_V\left[ \|BX(X^\top VX - I_p) - BX(X^\top BX - I_p)\|^2 \right]. \tag{82}$$

The first term in the above is upper bounded as

$$\mathbb{E}_{U,V}\left[ \|UX(X^\top VX - I_p) - BX(X^\top VX - I_p)\|^2 \right] \leq \mathbb{E}_{U,V}\left[ \|U - B\|_\text{F}^2 \|X(X^\top VX - I_p)\|_2^2 \right] \tag{83}$$
$$= \sigma_B^2 \mathbb{E}_V[\|X(X^\top VX - I_p)\|_2^2] \tag{84}$$

and the second is controlled by

$$\mathbb{E}_V\left[ \|BX(X^\top VX - I_p) - BX(X^\top BX - I_p)\|^2 \right] = \mathbb{E}_V\left[ \|BXX^\top(V - B)X\|^2 \right] \tag{85}$$
$$\leq \sigma_B^2 \|B\|_2^2 \|X\|_2^6. \tag{86}$$

Taking things together

$$\text{Var}\left[ UX(X^\top VX - I_p) \right] \leq (\mathbb{E}_V[\|X(X^\top VX - I_p)\|_2^2] + \|B\|_2^2 \|X\|_2^6)\sigma_B^2 \tag{87}$$
$$\leq \left( \frac{1+\varepsilon}{\beta_n}\mathbb{E}_V[\|X^\top VX - I_p\|_2^2] + \frac{(1+\varepsilon)^3}{\beta_n} \right)\sigma_B^2, \tag{88}$$

where for the second inequality we can use the bounds on the singular values of $X \in \mathrm{St}_B^\varepsilon(p, n)$.

Similarly, the variance of the first term in the landing is controlled by introducing yet another random variable $G$ that takes values from $\nabla f_\xi(X)$. We use the U-statistics variance decomposition twice to get

$$\mathrm{Var}[\mathrm{skew}\left(GX^\top U\right)VX] = \mathbb{E}_{G,U,V}[\|\mathrm{skew}((G - \nabla f(X))X^\top U)VX\|^2] \tag{89}$$

$$+ \mathbb{E}_{U,V}[\|\mathrm{skew}(\nabla f(X)X^\top(U - B))VX\|^2] \tag{90}$$

$$+ \mathbb{E}_V[\|\mathrm{skew}(\nabla f(X)X^\top B)(V - B)X\|^2] \tag{91}$$

which leads to the bound

$$\mathrm{Var}[\mathrm{skew}\left(GX^\top U\right)VX] \leq \sigma_G^2 \mathbb{E}_U[\|UX\|_2^2]^2 + \sigma_B^2(\|\nabla f(X)X^\top\|_2^2 \mathbb{E}_U[\|UX\|_2^2] + \|\nabla f(X)X^\top B\|_2^2 \|X\|_2^2) \tag{92}$$

Joining the two bounds above, we get the result. $\qquad\square$

### D.5 PROOF OF PROPOSITION 3.4

*Proof.* Same as in the proof of Theorem 2.8, by telescopically summing and averaging the iterates in (54), we arrive at the inequality

$$\frac{\eta\rho}{4K} \sum_{k=1}^K \|\Psi_B(X^k)\|^2 + \frac{\eta\rho\omega^2}{4K} \sum_{k=0}^K \|h(X^k)\|^2 \leq \frac{\mathcal{L}(X^0) - \mathcal{L}(X^{K+1})}{K} + \frac{L_\mathcal{L}\eta^2\gamma^2}{2}, \tag{93}$$

which implies also that the following two inequalities hold individually

$$\frac{1}{K} \sum_{k=1}^K \|\Psi_B(X^k)\|^2 \leq \frac{2}{\rho}\left(2\frac{\mathcal{L}(X^0) - \mathcal{L}(X^{K+1})}{K\eta} + L_\mathcal{L}\eta\gamma^2\right) \tag{94}$$

$$\frac{1}{K} \sum_{k=0}^K \|h(X^k)\|^2 \leq \frac{2}{\rho\omega^2}\left(2\frac{\mathcal{L}(X^0) - \mathcal{L}(X^{K+1})}{K\eta} + L_\mathcal{L}\eta\gamma^2\right). \tag{95}$$

In the above we see that the optimal step-size given $K$ iterations is

$$\eta^* = \frac{\sqrt{2(L(X^{K+1}) - L(X^0))}}{\sqrt{K L_\mathcal{L}}\gamma} \tag{96}$$

and the value of the parenthesis on the right-hand side becomes $2\sqrt{2(\mathcal{L}(X^0) - \mathcal{L}(X^K))L_\mathcal{L}/K}\gamma$. We thus need

$$K = 32 L_\mathcal{L}\gamma^2 \frac{\mathcal{L}(X^0) - \mathcal{L}(X^K)}{e^2\rho^2} \tag{97}$$

iterations to decrease $\inf_{k \leq K} \mathbb{E}\|\Psi(X^k)\| \leq e$ and similarly, but with extra $\omega^4$ in the denominator, for the constraint $\inf_{k \leq K} \mathbb{E}\|h(X^k)\| \leq e$.

Consider batch $r = 1$, since each iteration cost $npr$, we have the following number flops to get $e$-critical point

$$32 L_\mathcal{L}\gamma^2 \frac{\mathcal{L}(X^0) - \mathcal{L}(X^K)}{e^2\rho^2} np. \tag{98}$$

Taking that $L_\mathcal{L} = \mathcal{O}(\alpha_1 + \beta_1 + \kappa)$ from the previous Lemma D.1 and by the fact that $\rho = \mathcal{O}(1/\kappa)$, we have that we require $\mathcal{O}(\gamma^2\kappa^2(\kappa + \alpha_1 + \beta_1)np/e^2)$ flops.

It remains to estimate the variance of the landing $\gamma$ in terms of the variances of $\sigma_G, \sigma_B$ using Proposition 3.3, which states:

$$\gamma^2 \leq \sigma_G^2 p_B^2 \frac{(1+\varepsilon)^2}{\beta_n^2} + \sigma_B^2 \frac{1+\varepsilon}{\beta_n}\left(4\Delta(p_B + \beta_1^2) + p_N + (1+\varepsilon)^2\right). \tag{99}$$

Here we have that $p_N = \frac{1+\varepsilon}{\beta_n}\sigma_B^2 + \varepsilon$, $\Delta = \sup_{X \in \mathrm{St}_B^\varepsilon(p,n)} \|\nabla f(X)X^\top\|_2^2$, and $p_B = \mathbb{E}_\zeta\|B_\zeta\|_2^2$ which can be bounded as $p_B \leq \beta_1^2 + \sigma_B^2$. When the variance of the constraint is small and we have that $\sigma_B < \beta_1$ we get $p_B \leq 2\beta_1^2$ and

$$\gamma^2 \leq 8\kappa^2\beta_1^2\sigma_G^2 + (24\beta_1\kappa + 10/\beta_n)\sigma_B^2 + \frac{4}{\beta_n}\sigma_B^4 \tag{100}$$

where we also use that $\varepsilon < 1$. This gives an asymptotic bound

$$\gamma^2 \leq \mathcal{O}\left(\kappa^2\beta_1^2\sigma_G^2 + \left(\beta_1\kappa + \beta_n^{-1}\right)\sigma_B^2 + \sigma_B^4/\beta_n\right), \tag{101}$$

leading to the asymptotic number of floating point operations for $e$-criticality to be

$$\left(\kappa^2\beta_1^2\sigma_G^2 + \left(\beta_1\kappa + \beta_n^{-1}\right)\sigma_B^2 + \sigma_B^4/\beta_n\right)\frac{\kappa^2(\kappa + \alpha_1 + \beta_1)np}{e^2}, \tag{102}$$

where the leading term is

$$\left(\kappa\beta_1\sigma_G^2 + \left(1 + \beta_1^{-2}\right)\sigma_B^2\right)\frac{\beta_1\kappa^3(\kappa + \alpha_1 + \beta_1)np}{e^2}. \tag{103}$$

$\square$

# E    RIEMANNIAN INTERPRETATION OF $\Psi_B^{\mathrm{R}}(X)$ IN PROP. 3.2

Similar to the work of (Gao et al., 2022b), we can provide a geometric interpretation of the relative descent direction $\Psi_B^{\mathrm{R}}(X)$ as a Riemannian gradient in a canonical-induced metric and the isometry between the standard Stiefel manifold $\mathrm{St}(p,n)$ and the generalized Stiefel manifold $\mathrm{St}_B(p,n)$. Let

$$\mathrm{St}_{B,M}(p,n) := \{X : X^\top B X = M\},$$

for $B, M \succ 0$, which is the sheet manifold of $\mathrm{St}_B(p,n)$, and consider a map

$$\Phi_{B,M} : \mathrm{St}(p,n) \to \mathrm{St}_{B,M}(p,n) : Y \mapsto B^{-\frac{1}{2}}YM^{\frac{1}{2}}.$$

The map $\Phi_{B,M}$ acts as a diffeomorphism of the set of the full rank $\mathbb{R}^{n\times p}$ matrices onto itself and maps the standard Stiefel manifold $\mathrm{St}(p,n)$ to the generalized Stiefel manifold $\mathrm{St}_{B,M}(p,n)$. It is easy to obtain the tangent space at $X \in \mathrm{St}_{B,M}(p,n)$ via the standard definition:

$$
\begin{aligned}
\mathrm{T}_X\mathrm{St}_{B,M}(p,n) &= \{\xi \in \mathbb{R}^{n\times p} : \xi^T B X + X^T B \xi = 0\} \\
&= \{X(X^T B X)^{-1}\Omega + B^{-1}X_\perp K : \Omega^T + \Omega = 0, \Omega \in \mathbb{R}^{p\times p}, K \in \mathbb{R}^{(n-p)\times p}\} \\
&= \{WBX : W^T + W = 0, W \in \mathbb{R}^{n\times n}\} \\
&= \{\Phi_{B,M}(\zeta) : \zeta \in \mathrm{T}_{\Phi_{B,M}^{-1}(X)}\mathrm{St}(p,n)\}
\end{aligned}
$$

Consider the canonical metric on the standard Stiefel manifold $\mathrm{St}(p,n)$:

$$g_Y^{\mathrm{St}(p,n)}(Z_1, Z_2) = \left\langle Z_1, (I - \frac{1}{2}YY^T)Z_2 \right\rangle$$

Using the map $\Phi_{B,M}$, we define the metric $g^{\mathrm{St}_{B,M}(p,n)}$ which makes $\Phi_{B,M}$ an isometry as. Hence, we have that this metric is defined as

$$
\begin{aligned}
g_X^{\mathrm{St}_{B,M}(p,n)}(\xi, \zeta) &= g_{\Phi_B^{-1}(X)}^{\mathrm{St}(p,n)}(\Phi_B^{-1}(\xi), \Phi_B^{-1}(\zeta)) \\
&= \left\langle \xi, (B - \frac{1}{2}BX(X^T B X)^{-1}X^T B)\zeta(X^T B X)^{-1} \right\rangle.
\end{aligned}
$$

Consequently, the corresponding normal space of $\mathrm{St}_{B,M}(p,n)$ is

$$\mathrm{N}_X\mathrm{St}_{B,M}(p,n) = \{X(X^T B X)^{-1}S : S^T = S, S \in \mathbb{R}^{p\times p}\}.$$

The form of the derived tangent and normal spaces allow us to derive their projection operators $P_X$ and $P_X^\perp$ respectively as

$$
\begin{aligned}
P_X^\perp(Y) &= X(X^T B X)^{-1}\mathrm{sym}(X^T B Y), \\
P_X(Y) &= Y - X(X^T B X)^{-1}\mathrm{sym}(X^T B Y).
\end{aligned}
$$

Since $\Phi_{B,M}$ is isometric, the Riemannian gradient w.r.t. $g^{\mathrm{St}_{B,M}(p,n)}$ can be computed directly by

$$
\begin{aligned}
\mathrm{grad}_{B,M} f(X) &= \Phi_{B,M}(\mathrm{grad}_Y f(Y)) \\
&= \Phi_{B,M}(\mathrm{grad}_{\Phi_{B,M}^{-1}(X)} f(\Phi_{B,M}^{-1}(X))) \\
&= B^{-\frac{1}{2}} \mathrm{grad}_{\Phi_{B,M}^{-1}(X)} f(\Phi_{B,M}^{-1}(X)) M^{\frac{1}{2}} \\
&= 2 B^{-\frac{1}{2}} \mathrm{skew}\left(\nabla f(\Phi_{B,M}^{-1}(X))(\Phi_{B,M}^{-1}(X))^T\right) \Phi_{B,M}^{-1}(X) M^{\frac{1}{2}} \\
&= 2\mathrm{skew}(B^{-\frac{1}{2}} \nabla f(B^{\frac{1}{2}} X M^{-\frac{1}{2}}) M^{-\frac{1}{2}} X^T) BX \\
&= 2\mathrm{skew}(B^{-\frac{1}{2}} B^{-\frac{1}{2}} \nabla f(X) M^{\frac{1}{2}} M^{-\frac{1}{2}} X^T) BX \\
&= 2\mathrm{skew}(B^{-1} \nabla f(X) X^T) BX.
\end{aligned}
$$

Hence, akin to the work of (Gao et al., 2022b) for the standard Stiefel manifold, we derived the equivalent Riemannian interpretation of $\Psi_B^{\mathrm{R}}(X)$ and the landing algorithm for the generalized Stiefel manifold $\mathrm{St}_B(p,n)$. Note, the formula for $\Psi_B^{\mathrm{R}}(X)$ involves computing an inverse of $B$ and thus does not allow a simple unbiased estimator to be used in the stochastic case, as opposed to $\Psi_B(X)$.