# OpenReview forum: "Optimization without retraction on the random generalized Stiefel manifold for canonical correlation analysis"
_ICLR.cc/2024/Conference — Submitted to ICLR 2024_

### Official Review · Reviewer_6Nz4 · 2023-10-26

**Soundness:** 4 excellent
**Presentation:** 4 excellent
**Contribution:** 4 excellent
**Rating:** 10
**Confidence:** 4

**Summary:**

This paper presents a landing method for optimization over the generalized Stiefel manifold. The proposed method is inexpensive compared to classical optimization on Riemanian manifold. Furthermore the authors propose a stochastic version of the algorithm leading to an inexpensive and scalable method. The experiments show the good quality of the framework for different machine learning tasks.

**Strengths:**

This paper presents a general framework for optimization over the generalized Stiefel manifold. Such framework have many possible applications in the machine learning world (e.g. CCA, SVD...). The great strength of the method is its computational cost compared to classical framework for such optimization. The stochastic version is a great addition for scalability.

**Weaknesses:**

The framework has few weakness. The sublinear convergence shows that getting a suitable estimate may ask for many iterations as shown in the experiments. If the quality of the estimation is crucial the computation, even if inexpensive, can be long.

**Questions:**

I have one minor remark, on figure 3(b) of the left graphics the objective value followed the time when all others follow the iterations.

---

> ### Author Response · Authors · 2023-11-20
> **Response to the Reviewer 6Nz4**
>
> We thank the reviewer for being so enthusiastic about our paper and recognizing its strengths, this is extremely encouraging!
>
> We agree that sublinear convergence is a potential weakness of the method, which we diligently acknowledge in the paper, but we do not feel that this can be overcome since even Riemannian SGD has such a global convergence rate on a non-convex problem (Boumal et. al, 2019).
>
> We fixed the incorrectly labeled x-axis in Figure 3.

---

> > ### Comment · Reviewer_6Nz4 · 2023-11-22
> > **Global response to the authors**
> >
> > I read the different answers of the authors. Many thanks for the different clarification.

---

### Official Review · Reviewer_44jd · 2023-10-30

**Soundness:** 2 fair
**Presentation:** 3 good
**Contribution:** 2 fair
**Rating:** 5
**Confidence:** 3

**Summary:**

Optimization under generalized orthogonality constraints captures many applications in machine learning. This paper extends Ablin & Peyré's landing method, adapting it for stochastic settings to address problems involving generalized orthogonality constraints.

The proposed method does not rigorously impose the constraint in each iteration but rather produces a series of iterations that gradually conform to the generalized Stiefel manifold. Therefore, it can avoid the retraction step. They show that the proposed method is convergent, and any clustering point is a critical point. The authors have conducted experiments to validate the efficacy of the proposed methods.

**Strengths:**

S1. A retraction-free algorithm is introduced to solve the optimization problem involving generalized orthogonality constraints.

S2. The authors demonstrate that the proposed method converges to a critical point of Equation (1) in both deterministic and stochastic cases.

S3. They provide numerical evidence of the proposed method's efficiency through the deterministic generalized eigenvalue problem and through the stochastic CCA.

**Weaknesses:**

W1. The adaptation of Ablin & Peyré's landing method to accommodate the general setting with $\mathbf{B}\neq \mathbf{I}$ and the stochastic context is straightforward.

W2. The convergence results are relatively weak. These results could be readily derived using the sufficient descent condition.

W3. The time complexity comparison in Table 1 is biased, and the motivation for the retraction-free method is not strong. The authors argue that the polar method has a time complexity of $O(n^2 p)$, while the proposed method can leverage the rank-r structure of matrix $\mathbf{B}$ and achieve a time complexity of $O(\min(n^2 p, nrp))$. However, if the polar method takes advantage of the rank-r structure of matrix $\mathbf{B}$ for matrix multiplication, its complexity could also be greatly reduced when employing the Woodbury matrix identity.

W4. The paper lacks comparisons with the Multiplier Correction Methods (Gao et al., 2019a, 2022a). The authors only briefly mention that the Multiplier Correction Methods are sensitive to the appropriate selection of the penalty parameter, but a direct comparison with these methods is necessary.

W5. The authors should compare their method against the modified Gram-Schmidt-based generalized polar decomposition method (e.g., "A practical Riemannian algorithm for computing dominant generalized Eigenspace, Z Xu, P Li, UAI 2020").

W6. The parameter $\omega$ is crucial for the convergence of the algorithm. The authors should demonstrate how this parameter's sensitivity affects the performance of the algorithm.

**Questions:**

Is the inequality in Equation (11) tight? What are the values for the upper bound (namely $\eta(x)$) in practice? I suspect that restricting the step size could greatly limit the practical efficacy of the proposed algorithm.

---

> ### Author Response · Authors · 2023-11-20
> **Response to the Reviewer 44jd (1 of 3)**
>
> We thank the reviewer for their time reading our paper and their comments.
>
>  **W1, W2**: We find it important to address the first two strong reviewer’s remarks, stating the results are “straightforward”, “can be readily derived using sufficient descent condition” and are “relatively weak”.
> - Stating that results are “straightforward” is a **subjective statement**, for example, it clashes with **the opinion of reviewer 1p92** stating the results are **“important and non-trivial”**. Nevertheless, generally in science, the simplicity of a result does not determine its quality or significance.
> - The objective fact is that, as reviewed in Section 1.1, our results cannot be derived using existing theory, in particular: i) Riemannian optimization techniques employing retractions do not allow for optimization over randomly changing manifolds, since the projection on a random constraint does not yield an unbiased estimator. ii) Infeasible optimization techniques such as the mentioned multiplier correction methods (Gao et al. 2019a, 2022a), also reviewed in Section 1.1, do not consider stochastic constraints and lack stochastic convergence results. We give further detail into the difference in the proof technique at the start of Sec 2 and also at the bottom of this reply*.
> - “relative weakness of the result”, could the reviewer please elaborate: relative to what? As stated in the introduction, the landing method: achieves **the same best known global asymptotic sublinear convergence rate** as its Riemannian counterpart for the general smooth non-convex deterministic objective in eq (1), see “Global rates of convergence for nonconvex optimization on manifolds“ (Boumal et al., 2019), while having a cheaper per-iteration complexity and also solving the stochasticly constrained problem in expectation. In particular, **for the stochastic constrained case, this is the first time** a method is proposed for $X^\top B X = I$ equality-constrained optimization that allows for noise in the normal space while ensuring convergence in expectation. We are not aware of an equivalent existing result for a general smooth non-convex objective over the random generalized Stiefel manifold.
> - “These results could be readily derived using the sufficient descent condition.”, could the reviewer please explain what they mean by this statement: sufficient descent condition on which quantity? If the reviewer has in mind the merit function that we introduce, then the reviewer is correct, but could then say the exact same thing about all optimization papers that use a merit function, and the reviewer would dismiss as trivial the majority of the literature. The centerpiece of our theoretical analysis is to build a novel merit function that provably decreases, but the hard part here is to design such a function. The proposed merit function is novel and allows us to prove results in harder settings than previously proposed methods like (Schechtmann et al 2022). For instance, we can still guarantee convergence when there is noise in the normal space while previous analysis techniques fail here.
>
> (continues in the next 2/3 part)
>
> (* Note, the proof technique used in the previous lines of work for the orthogonal manifold in (Schechtman et al., 2022) and in (Ablin et. al., 2023) does not readily apply here. The technique in (Schechtman et al., 2022) fails due to the non-smooth merit function not allowing to show decrease in expectation for the distance to the constraint and the technique in (Ablin et. al., 2023) does not provide a general way for constructing a smooth merit function.)

---

> ### Author Response · Authors · 2023-11-20
> **Response to the Reviewer 44jd (2 of 3)**
>
> (continued from 1/3 part)
>
>  **W3**: The difference between the per-iteration computation cost of our landing method compared to other methods is crucial and we want to clarify certain points:
> - We believe the confusion of the time-complexity comparison arose due to comparing stochastic and deterministic variants together. To clarify this difference, we split Table 1 to have two columns.
> - We want to stress that unlike what the reviewer writes, $B$ is **not low rank**, it is only the expectation of low-rank matrices (e.g. individual covariances of samples).
> - We have to disagree with the reviewer’s statement that a Woodbury identity can be used for fast low-rank updates of the polar retraction. The Woodbury identity $(I+UV)^{-1} = I + U(VU)^{-1}V$ **cannot** be used to get a closed form expression for a low-rank update of the matrix square root in the polar retraction; see the recent work “Low-rank updates of matrix square roots” (Shmueli et al., 2023) which shows that this requires solving a low-rank algebraic Riccati equation by solving a Riemannian optimization problem in itself. Furthermore, in the polar retraction $(X+Z)(I_p +Z^\top BZ)^{-1/2}$, if $B$ is updated with a low-rank update, the matrix inverse square root $(I_p + Z^\top BZ)^{-1/2}\in\mathbb{R}^{p\times p}$ generally won’t have rank-$r$, thus multiplication from the left with $(X+Z)\in\mathbb{R}^{n\times p}$ will have the time complexity of $\mathcal{O}(n^2 p)$. However, even if it were possible to compute the polar retraction cheaper for the subsampled rank-r case, which it is not, the exact projection on the randomly sampled rank-$r$ constraint would not give convergence in expectation in the stochastic case. In contrast, our landing method has per-iteration complexity on the order $\mathcal{O}(npr)$ when $B$ is given as a sum of rank-r matrices and converges to the constraint in expectation.
> - Finally, as explained in Sec 1.1, another important practical asset of our method is that it only uses matrix multiplications, while retractions perform a projection on the random constraint, which requires **non-parallelizable matrix factorizations**, as shown in Table 1, and does not give an unbiased estimate in the stochastic case. Thus, even in the extreme case $n=p=r$ where the time complexity is the same, in practice, **matrix multiplication is always much faster**. We highlight this in the text and reference the new Figure 6b in the appendix, which shows this experimentally.
> - Overall, our experiments clearly demonstrate that the proposed method is practically faster on GPU hardware than classical Riemannian approaches while requiring only $\mathcal{O}(np)$ memory in the online setting.
>
> W4: The reviewer asks for a more detailed comparison with multiplier correction methods
> - We added a comparison with PLAM (Gao et al., 2022a) in the deterministic experiment shown in Fig. 2
> - We want to highlight that methods, such as PLAM (2019a, Gao et al., 2022a) reviewed in Sec 1.1, contrary to our work, are only deterministic and do not have convergence guarantees in expectation when $B$ or $\nabla f(X)$ are randomly sampled, which is one of our core contributions. To clarify this vital point, we added a new first sentence in Sec 1.2. (Comparison with the landing method).
> - To support the claim that, in the deterministic case, the multiplier correction method PLAM is more sensitive to the parameter $\omega$, we: (i) added a reference to a sensitivity experiment for the standard Stiefel manifold in (Fig 9, Ablin & Peyre, 2022), which shows that PLAM can diverge when $\omega$ is too large, even if the step-size is very small, and (ii) added the new Fig. 7 in the appendices showing convergence behaviour for varying choices for the step-size $\eta$ and the parameter $\omega$ for PLAM and our landing method.
>
> (continues in the next 2/3 part)

---

> ### Author Response · Authors · 2023-11-20
> **Response to the Reviewer 44jd (3 of 3)**
>
> (continued from 2/3 part)
>
>  **W5**: The reviewer asks specifically for a comparison with a generalized eigenvalue solver in ("A practical Riemannian algorithm for computing dominant generalized Eigenspace, Z Xu, P Li, UAI 2020", https://proceedings.mlr.press/v124/xu20a.html):
> - We added the requested algorithm  “rgGenELinK”/”rgCCALin” to the summary Sec 1.1 and Table 2 to provide easy comparison of complexities, but since there is no public code for the algorithm and the algorithm is neither stochastic nor solves the general smooth non-convex objective problem in eq (1) , we did not reimplement the method to provide numerical comparisons. The difference between GEVP solvers and our method is also highlighted in Section 1.2 (Comparison with the landing method)
>
>  **W6**: To provide more details about the sensitivity of our landing method to the parameter $\omega$. We:
> - added a reference to a sensitivity experiment for the standard Stiefel manifold in (Fig 9, Ablin & Peyre, 2022), and
> - added new Fig 7 and Fig 9 in the appendices showing the convergence behaviour  of the landing method for varying choices for the step-size $\eta$ and the parameter $\omega$.
>
>  **Q**: The safe-step size bound in Lemma 2.3 is not tight. To show that the upper bound is only mildly restricting:
> - we want to highlight the result of Lemma 2.4 which shows that it remains bounded away from zero.
> - we can track the upper-bound in eq (11) numerically through iterations and we plot them in the new Fig 8 in the appendices showing that the upper bound is only mildly restricting at the start of the iterations and becomes relaxed as the algorithm approaches a stationary point. In practice we employ the standard approach used for stochastic gradient algorithms: we look for a suitable fixed step-size using grid search.
>
>  **Flag of ethics review**: We are unsure why the reviewer flags this paper for ethics review, especially without providing any form of explanation. Our results are mostly theoretical.
>
> **Concluding remarks** Please respond to our post to let us know if the above clarifications adequately address your concerns about our work. We are happy to address any remaining points during the discussion phase; if the above responses are sufficient, we kindly ask you to consider raising your rating.

---

### Official Review · Reviewer_dA73 · 2023-11-01

**Soundness:** 3 good
**Presentation:** 2 fair
**Contribution:** 3 good
**Rating:** 5
**Confidence:** 3

**Summary:**

This paper delves into the optimization problem associated with the generalized Stiefel manifold, employing methods that do not involve retraction. The authors introduce a novel 'landing' algorithm, extending prior work by Ablin and Peyre (2022). Notably, the landing algorithm offers the advantage of efficient iteration computation complexity. The study investigates the convergence properties of both deterministic and stochastic algorithms when applied to the generalized Stiefel manifold.

**Strengths:**

This paper generalizes previous landing algorithms to stochastic setting for both constraint and objective. The convergence results are given for deterministic and stochastic cases.

**Weaknesses:**

1. The paper lacks a clear explanation regarding the rationale behind the use of two random variables, $\zeta$ and $\zeta'$, as mentioned in equation (3). While the paper briefly mentions that noise in the tangent space is allowed and that the relative gradient has an unbiased estimate, it fails to adequately address how this difficulty is overcome in the main context of the research. It would be beneficial to provide more clarity on this aspect in the main body of the paper.

2. Table 2 reveals that previous works have enjoyed a linear convergence rate, whereas this work achieves a sublinear rate. It is essential to acknowledge this difference in convergence rates. Furthermore, the paper's proved results appear to be more favorable when the condition number of matrix $B$ is low, and this should be explicitly mentioned.

**Questions:**

1. In the introduction, the paper stipulates that $B_{\zeta}$ must be positive definite. However, this requirement implies that the sample size should exceed the dimensionality $n$. Could the authors clarify why this is a necessary condition and how it relates to the sample size?
2.  Could the paper provide insight into the derivation of the distinct formulas for $\Phi_B(X)$ and $\Phi_B^R(x)$ as presented in Proposition 3.2? Are these formulas related to different metrics? Additionally, in Figure 2, the term 'landing (Riem. gradient)' is used. It seems that "Riemannian steepest descent with QR-based Cholesky retraction (Sato & Aihara, 2019)" is a retraction-based method. Could the paper explain why the distance $\mathcal{N}(x)$ is not zero in this context?

---

> ### Author Response · Authors · 2023-11-20
> **Response to the Reviewer dA73 (1 of 2)**
>
> We would like to thank the reviewer for carefully reading our work, recognizing its strengths as **“giving stochastic and deterministic convergence results”** and having **“efficient iteration computation complexity”**. We are also thankful for their additional questions about the core concepts of the work, which we wish to clarify
>
>  **W1**: There are two parts to the reviewer's remark: (i) unbiasedness of the normalizing estimator $\nabla N(X)$, and (ii) how we managed to extend the theory to allow for noise to be in the normal space as opposite to (Schechtman et al. 2022):
> - i) Just like classical theory requires stochastic directions in SGD to be unbiased, we need to have unbiased estimates of the landing direction. Therefore, we need to find an unbiased estimator of the normalizing component, $BX(X^T B X - I)$. The simplest way to ensure the unbiasedness of a quadratic function of an expectation is to sample two *independent* matrices $B_\zeta$ and $B_{\zeta’}$ in the landing formula in eq (3), which leads to the stochastic estimator that we have considered. We are not aware of the existence of another way to ensure its unbiasedness using the theory of U-statistics (Van der Vaart, 2000). Note that this is only a minor technical hindrance and not a practical problem: when $B$ is a randomly sampled covariance matrix, we can simply use two independently sampled mini-batches to construct two independent estimates of the covariance B.
> - ii)The other issue is allowing for noise in the normal space. First, we believe this is an important problem since it naturally arises in the generalized Stiefel manifold studied here. We want to highlight the beginning of Sec 2., where we introduce our method and the main ideas that overcome this difficulty. Previous analysis technique of (Schechtman et al. 2022) does not work in this context because of the non-smoothness of the cost function they use, which is why we had to explore a novel avenue of proof based on a smooth cost function. This allowed us to prove convergence of our method.
>
>  **W2**: The reviewer is correct in that the methods in Table 2 have different convergence rates to an e-critical point that must be acknowledged. We want to highlight that we already display this clearly in Table 2: the deterministic solvers for the generalized eigenvalue problem and CCA achieve linear convergence $\mathcal{O}(\log(1/e))$ whereas the stochastic solvers achieves sublinear $\mathcal{O}(1/e^2)$ convergence and also in the footnote at the bottom of page 5. For the sake of clarity, _we have now stated this difference clearly verbatim in the caption of Table 2_. We also want to highlight the Sec. 1.2 (Comparison with the landing method) which states that the landing method: (i) outperforms the reviewed GEVP solvers only for well-conditioned matrices, and also (ii) that it is designed to solve any **general non-convex optimization problem** in eq (1) (so not only CCA and GEVP) for which the best known deterministic convergence rates are sublinear, and to our knowledge, do not exist for the stochastic constrained case. For greater clarity, _we specified that well-conditioned means that $\kappa$ is small_.
>
>  **Q1**: We want to clarify that the sampled $B_\zeta$ does not need to be positive definite but only positive definite in expectation.  In other words, we only need that each $B_\zeta\succcurlyeq0$, and that on average $E[B_\zeta] \succ 0$. As such, there is **no** condition on $r\geq n$, this is related to the fact that our landing method **does not project** exactly, but instead, only **moves with an unbiased stochastic estimator, towards** the constraint. We have clarified this extremely important point in the text, which was unclear in the first version. We added a sentence in the third introductory paragraph and we changed the first sentence of the paper by adding $E[B_\zeta] \succ 0$ highlighted in red.
>
>  **Q2**: The reviewer points out an interesting question about what is the insight into how the two formulas for $\Psi_B(X)$ and $\Psi^R_B(X)$ in Prop 3.2. were derived and what their meaning is:
> - It is possible to express $\Psi^R_B(X)$ as a Riemannian gradient with a specifically chosen metric. This is in line with a similar work done in (Ablin et al., 2023). We added a remark after Prop 3.2 referencing the newly added Appendix E that derives $\Psi^R_B(X)$.
> - The geometric interpretation of the unbiased formula $\Psi_B(X)$ is less clear. We use the fact that for the case of a generalized Stiefel manifold there exists a closed form formula for an oblique projection on the tangent space that is linear in $B$ and $\nabla f(X)$.
>
> (continues in the next 2/2 part)

---

> > ### Comment · Reviewer_dA73 · 2023-11-22
> >
> > In the convergence analysis, the authors draw a comparison with the technique from Schechtman et al. (2022). However, it appears that Fletcher's augmented Lagrangian function, defined in equation (9) of this paper, is similar to the merit function in equation (16) from Ablin et al. (2023). The proof sketches in both works also show similarities, mainly differing in the handling of stochastic constraints. This raises concerns about the level of novelty and contribution of the current paper. It would be helpful if the authors could clarify how their approach diverges from the existing literature and emphasize the unique elements of their method.
> >
> >
> >
> > Pierre Ablin, Simon Vary, Bin Gao, and P-A Absil. Infeasible Deterministic, Stochastic, and
> > Variance-Reduction Algorithms for Optimization under Orthogonality Constraints. arXiv preprint
> > arXiv:2303.16510, 2023.

---

> ### Author Response · Authors · 2023-11-20
> **Response to the Reviewer dA73 (2 of 2)**
>
> (continued from 1/2 part)
>
> Minor clarity question about Fig 2 and Fig 3:
> - There was a confusion regarding why in Fig. 2, the method “Landing (Riem. gradient)” did not achieve exactly $\mathcal{N}(X) = 0$. This is because this line corresponds to the landing method (eq. 2) using $\Psi^R_B$, that converges to the constraint in the limit, not the Riemannian gradient descent algorithm. We changed this confusing label to “Landing with $\Psi^R_B(X)$” to be in line with notation in Prop. 3.2, which hopefully clarifies the issue.
>
>
> **Concluding remarks** Please respond to our post to let us know if the above clarifications adequately address your concerns about our work. We are happy to address any remaining points during the discussion phase; if the above responses are sufficient, we kindly ask you to consider raising your rating.

---

> ### Author Response · Authors · 2023-11-22
>
> We thank the reviewer for reading our rebuttal and engaging in the discussion!
>
> We would like to take this opportunity and ask whether we managed to sufficiently address the other points raised in your original review?
>
> Key differences in terms of results with Ablin et al. (2023):
> 1. Our work describes a way to analyze landing algorithms for a general $h(x) = 0$
> 2. Our work allows for random constraints $X^\top B X = I_p$ allowing updates with noise in the normal space of the manifold.
> 3. Our work uses the exact Fletcher’s augmented Lagrangian, whereas Ablin et al. (2023) does not say how their merit function for $X^\top X = I_p$ is constructed
> 4. Our work, when considering $h(X) = X^\top X - I_p$ does not simplify to the merit function in Ablin et al. (2023)
> To our knowledge, there is no straightforward analogue of eq (9) in Ablin et al. (2023) which would derive the results for $X^\top B X = I_p$.
> 5. Our work gives a thorough analysis of the complexity bounds for GEVP and CCA which are competitive with the existing complexity bounds for **stochastic CCA** while being cheaper to compute.
>
> It is a very good question to ask whether the _proof technique_ of Ablin et al. (2023) for the optimization over $X^\top X = I_p$, which although, as the reviewer correctly points out, **does not consider a random constraint**, would be easily applicable in the case of $X^\top B X = I_p$ when $B$ is random.
>
> It is important to acknowledge that our proof technique based on a merit function in eq (9) is, at first sight, similar to the merit function in eq (16) of Ablin et al. (2023). However,
> * The work of Ablin et al. (2023) **does not give a general way** to define a merit function for a manifold constraint $h(x) = 0$ and it is not clear from their work how one would generalize eq (16) for $X^\top X = I$ to a general smooth manifold defined as $h(x)=0$
> * In the case when $h(X) = X^\top X-I_p$, i.e. the problem becomes the same as the one analyzed in Ablin et al. (2023), our merit function defined in eq (9) **is not the same** as the one defined in eq (16) of Ablin et al. (2023).
> * It would be sensible to try a straightforward adaptation of the merit function in eq (16) of Ablin et al. (2023) to the case of $X^\top B X = I_p$, e.g.:
> $$\mathcal{L}(X) = f(X) - \frac{1}{2}\langle \mathrm{sym}(X^\top B X), X^\top B X – I_p \rangle + \mu \||X^\top B X – I_p \||^2_F,$$
> 	which looks at first as a viable generalization. However, **this formula does not work** for $X^\top B X = I_p$, to be specific in the proof of Ablin et al. (2023) in eq (62) on pg 25, one gets terms $X^\top B^2 X$. It is impossible to scale the middle term $\langle \cdot, \cdot\rangle$ term by a scalar to recover appropriate bounds. We also tried other reasonable candidates for $\mathcal{L}(X)$ based on Ablin et al. (2023) (and also Schechtman et al., (2023)) but it required the recent Riemannian insights from Goyens et al. (2023) about the **exact Fletcher’s augmented penalty function** in conjunction with the general algorithmic concept presented in Ablin et al. (2023) to get our result for a random constraint, and we hope, this makes our contribution novel enough.
> * The work of Ablin et al. (2023) does not include **analysis for CCA/GEVP**
> * The work of Ablin et al. (2023) does not allow for a **random constraint**, i.e. to have noise in the normal space of the manifold.
>
> We want to make these points about the used merit function clear and we added a brief explanation in Sec 1.2. (Comparison with the landing) highlighted in red:
>
> > Our work is conceptually related to the recently developed infeasible methods (Ablin & Peyre, 2022; Ablin et al., 2023; Schechtman et al., 2023), with the key difference of constructing a smooth merit function for a general constraint $h(x)$ that enables convergence analysis of iterative updates with error in the normal space of $\mathcal{M}$.

---

### Official Review · Reviewer_1p92 · 2023-11-02

**Soundness:** 3 good
**Presentation:** 3 good
**Contribution:** 3 good
**Rating:** 6
**Confidence:** 4

**Summary:**

In this paper the authors propose an optimization method in the (random) generalized Stiefel manifold, which doesn't require a retraction to the manifold. This work follows the line of some recent papers, which started by optimizing on the orthogonal group (Ablin and Peyré). All these formulations are based on a flow with two terms, one trying to minimize the function (or the norm of the gradient), and the other one minimizing the distance to the manifold. These terms are orthogonal to each other, under certain smart choices.

This paper contributes to the field in three ways: by considering a more general case for directions in the flow, by explicitly proposing a method for the generalized Stiefel manifold, and by addressing the random case.

The paper is well written in general. The theoretical part is sound. However, the experimental section is not convincing.

**Strengths:**

Out of the three contributions (more general descent directions, general Stiefel manifold, and random $B$), I think that the first one and the third one are the most important contributions.

The theoretical results are important and non-trivial.

**Weaknesses:**

To me, one of the strenghts is the application to the stochastic framework. However, the experimental section is particularly weak in that point. I don't know what to look for in Figure 3 (by the way, one label says "time", I assume incorrectly, and I don't know what "5 epochs" means in this context). The description in the text doesn't help much.

**Questions:**

In the paper there are results concerning feasible steps, bounds, and the existance of useful positive steps. How is the stepsize chosen in practice? At least in the experiments.

The choice $\Psi_B(X)$ seems to do a better job than $\Psi_B^R(X)$. Do you have any intuition about why that is?

I assume that the need for an inverse computation in $\Psi_B^R(X)$ makes this choice slower. What's the comparison in execution time?

Minor comments:

 - In Lemma 2.4, it reads "where $C_h$ is from Assumption 2.2", but $C_h$ is not present in the inequality. I think it should be $L_N$.

 - In the supplementary material, between equations (74) and (75), starting with the phrase "For the other choice of relative gradient ..." and to the end of the section, I think it would be better to use the same notation as in the paper, namely, $\Psi_B^R(X)$.

---

> ### Author Response · Authors · 2023-11-20
> **Response to the Reviewer 1p92**
>
> We thank the reviewer for their time and their thorough reading of the paper, for the recognition of our theoretical contribution as **“important and nontrivial”**, and for suggesting specific improvements to the text and the numerical experiments.
>
> **Comment about improving the numerical experiments:**
> > To me, one of the strenghts is the application to the stochastic framework. However, the experimental section is particularly weak in that point. I don't know what to look for in Figure 3 (by the way, one label says "time", I assume incorrectly, and I don't know what "5 epochs" means in this context). The description in the text doesn't help much.
>
> In the numerical experiments we fixed the problems with the existing Fig. 2 (deterministic setting) and Fig. 3 (stochastic setting) and added new experiments in Appendix B:
> - We corrected the legends in Fig 2 and Fig 3 with the correct notation $\Psi\_B(X)$ and $\Psi\_B^R(X)$ from Prop 3.2. We corrected the x-axis labels in Fig. 3.
> - We clarified in the text and the captions that each **epoch** corresponds to a pass through the whole dataset in the case of the stochastic CCA experiment.
> - We highlighted in the main text and the caption that the final experiment in Fig 3 corresponds to the “Stochastic Canonical correlation analysis” and illustrates the stochastic algorithm developed in the paper. We fixed the x-label to say “time (sec)” instead of iterations and there is Fig 5 in the appendices showing dependency on the iterations.
> - There is also a new discussion in the numerical section which highlights the difference of the stochastic methods in online setting, when each method has only few passes over the dataset.
>
> **Comment on the choice of the stepsize $\eta$ and the parameter $\omega$**
> > In the paper there are results concerning feasible steps, bounds, and the existance of useful positive steps. How is the stepsize chosen in practice? At least in the experiments.
>
> Thank you for the comment. We clarified in the main text that in practice the step-size can be selected by grid-search, as is standard for stochastic algorithms such as SGD, and we added the specific stepsizes used either in the main text, or referenced in the appendices due to lack of space. We also added new Fig 7 and Fig 9 showing the sensitivity of our method to the choice of $\omega$ and $\eta$ in the deterministic experiment.
>
> **Comments on the difference  between $\Psi_B(X)$ and $\Psi^R_B(X)$**
>
> > The choice $\Psi_B(X)$ seems to do a better job than $\Psi^R_B(X)$. Do you have any intuition about why that is?
>
> > I assume that the need for an inverse computation in $\Psi^R_B(X)$ makes this choice slower. What's the comparison in execution time?
>
> - We added Figure 6a in the appendix comparing the computation time of $\Psi_B(X)$, $\Psi^R_B(X)$, and standard Riemannian gradient, for the deterministic case when $n=p=r$, which is the most expensive case for $\Psi_B(X)$ and demonstrates the cost of computing the inverse in $\Psi^R_B(X)$ and the Riemannian gradient compared to using only the matrix multiplication in $\Psi_B(X)$. There is also new Fig 6b which gives a similar practical computation time comparison for retractions versus computation of the $\nabla \mathcal{N}(X)$. This highlights the correct intuition of the reviewer: the cost of computing $\Psi^R_B(X)$ is indeed significantly greater than that of  $\Psi_B(X)$.
> -  We clarified in the text that the fact that the landing algorithm $\Psi_B$ is converging faster compared to $\Psi^R_B(X)$ that requires the action of $B^{-1}$ in the timed comparison in Fig 2 is due to being able to perform more of cheaper iterations in the given time limit. Apart from the new Fig 6 comparing per-iteration computation time, we also added Fig 4. in the appendices, which is analogous to Fig 2, but tracks the number of iterations instead of time. Here we can see that in the given time limit of 2 minutes, the landing with $\Psi_B$ is able to perform almost twice as many iterations compared to the other methods.
> - We also added a new remark after Prop 3.2 that points to Sec. 5 in the appendices which gives a **geometrical interpretation** for $\Psi^R_B$ as a Riemannian gradient in a specific metric. Such interpretation is for the time missing for $\Psi_B$.
>
> **Minor comments:**
> - We corrected in the statement of Lemma 2.4.: “where $C_h$ is from Assumption 2.2” to “where $L_\mathcal{N}$ is from Assumption 2.2“.
> - In the supplementary material, we fixed the notation of $\Psi_B^R(X)$ to be consistent with that in the rest of the paper.
>
> **Concluding remarks** Please respond to our post to let us know if the above clarifications adequately address your concerns about our work. We are happy to address any remaining points during the discussion phase; if the above responses are sufficient, we kindly ask you to consider raising your rating.

---

> > ### Comment · Reviewer_1p92 · 2023-11-23
> >
> > I'm mostly satisfied with the authors' response, and the changes in the paper and supplementary material.
> > I'm still missing the intuition behind the better performance of $\Psi_B(X)$ (in iterations, the time complexity is well explained)

---

> ### Author Response · Authors · 2023-11-23
>
> We thank the reviewer for reading our rebuttal and we are glad we managed to address their concerns.
>
> The reviewer asks for a theoretical explanation why $\Psi_B (X)$ and $\Psi^\mathrm{R}_B (X)$ perform similarly well in terms of iterations (Fig. 4).
>
> This is a very good question, and the short answer is that this is an open question.
>
> As shown in Theorem 2.7, a key quantity to understand convergence speed is the constant $\rho$, which affects the upper bound on the convergence of $\||\Psi_B (X) \||^2_F$ _and_ also affects the optimal scaling of the normalizing $\omega$ parameter. However, it also affects the construction of the merit function $\mathcal{L}$ by having that $\beta = \mathcal{O}(\rho)$ in Lemma 2.6, which in turn could affect the numerator of the upper bound in Theorem 2.7. Moreover, it is important to point out that the convergence is given in terms of $\||\Psi_B (X) \||^2_F$ and that the property of Def 2.1 (ii) is not scale independent, meaning that taking $c\Psi_B (X)$ for $c>0$ will achieve $c\rho$. To highlight the point of Def 2.1 not being scale invariant we added a sentence under the Def 2.1. explaining this as:
> > Note, the above definition is not scale invariant to $\rho$, i.e. taking $c\Psi(x)$ for $c>0$ will result in $c\rho$, and this is in line with the forthcoming convergence guarantees deriving upper bound on $\||\Psi(x)\||$.
>
> The overall observed convergence in Fig. 4 will also depend on a **local convergence rate**, which could be derived by analysing an appropriate dynamical system around critical points and the spectral properties of its Jacobian. This is however beyond the scope of the present paper.
>
> The scope we envisioned for this work is to show the landing method is well suited for optimization over randomized constraints, a task very rarely tackled in optimization theory and important for representation learning, and derives a rare theoretical convergence result for fixed, sufficiently small step-sizes for any fixed $\omega>0$ over a **random constraint**. It is important for future work to have a better theoretical understanding of the convergence properties _and_ also of the optimal choices of the stepsize and $\omega$.

---

### Author Response · Authors · 2023-11-21
**Summary of the work and improvements**

We want to thank the reviewers for their detailed and encouraging responses. We have updated our paper according to their feedback, and the changes are highlighted in red.

To summarize our work, for the first time, a method is proposed for equality-constrained optimization that allows for noise in the normal space while ensuring convergence in expectation. This is a very natural goal, as it allows us to tackle constraints such as $X^\top BX = I$ where B is stochastic, with an important application in CCA. Yet no previous work, notably the closely related (Schechtman et al, 2022), was able to achieve this goal.

The main improvements on the paper are:
* We improved the discussion for the stochastic numerical experiment in Fig. 3 showing the stochastic CCA, and clarified the benefits of the landing method in the online setting per the recommendation of Reviewer 1p92
* We added a new Appendix E referenced after Prop 3.2, which gives a Riemannian interpretation to $\Psi^\mathrm{R}_B(X)$ in Prop. 3.2 to highlight the difference with $\Psi_B(X)$ as asked by Reviewer dA73 and Reviewer 1p92
* We added a new Fig. 6, which shows the per-iteration computation time comparison of the two landing methods compared to the Riemannian GD as pointed out by Reviewer 1p92. We also add a new Fig. 5 which is analogous to Fig. 2 but tracks number of iterations instead of time (sec.).
* We added in the deterministic numerical experiment shown in Fig. 2 an additional comparison with PLAM algorithm (Gao et al., 2022) and further highlighted that this method does not apply in the stochastic context in Sec 1.2 (Comparison with the landing) as suggested by Reviewer 44jd.
* We added a new experiment shown in Fig. 7 in Appendix B which demonstrates the landing method is more robust to the hyperparameter choice compared to the PLAM algorithm (Gao et al., 2022) as asked by Reviewer 44jd. We also reference in the numerical section an experiment from (Ablin & Peyre, 2022; Fig. 9) reaching the same conclusion for the standard Stiefel manifold.
* We added a new Fig. 8, which tracks numerically the value of the upper-bound on the safe step-size in Lemma 2.3, as asked by Reviewer 44jd.
* We added a new Fig. 9, which shows specifically how the step-size $\eta$ and $\omega$ effect the performance of the landing method as asked by Reviewer 44jd

---

### Meta-Review · Area_Chair_VUmb · 2023-12-16

**Metareview:**

This paper considers the problem of optimizing a function over the generalized Stiefel manifold, where both the objective function and the matrix defining the generalized Stiefel manifold can take an expectation form, and proposes a landing-type iterative method for solving it. While the extension of the landing method to the stochastic setting may be new, based on the feedback from the reviewers and my own reading of the paper, the contribution in this regard follows closely those in the literature and is at the borderline. Moreover, given that the paper aims to address a general class of optimization problems with generalized Stiefel manifold constraint, there should be more extensive numerical evaluation to demonstrate the superiority of the proposed method, including but not limited to comparing with other algorithms for the CCA problem. Thus, the paper can benefit from a more careful revision.

**Justification For Why Not Higher Score:**

The paper in its current form is marginal in terms of contributions and numerical validation.

**Justification For Why Not Lower Score:**

N/A

---

### Decision · Program_Chairs · 2024-01-16

Reject